# Spike and slab variational Bayes for high dimensional logistic regression

**Kolyan Ray**[*]
Department of Mathematics
Imperial College London
kolyan.ray@ic.ac.uk

**Botond Szabó**[*]
Department of Mathematics
Vrije Universiteit Amsterdam
b.t.szabo@vu.nl

**Gabriel Clara**
Department of Mathematics
Vrije Universiteit Amsterdam
g.clara@student.vu.nl

## Abstract

Variational Bayes (VB) is a popular scalable alternative to Markov chain Monte Carlo for Bayesian inference. We study a mean-field spike and slab VB approximation of widely used Bayesian model selection priors in sparse high-dimensional logistic regression. We provide non-asymptotic theoretical guarantees for the VB posterior in both $\ell_2$ and prediction loss for a sparse truth, giving optimal (minimax) convergence rates. Since the VB algorithm does not depend on the unknown truth to achieve optimality, our results shed light on effective prior choices. We confirm the improved performance of our VB algorithm over common sparse VB approaches in a numerical study.

## 1   Introduction

Let $x \in \mathbb{R}^p$ denote a feature vector and $Y \in \{0, 1\}$ an associated binary label to be predicted. In logistic regression, one of the most widely used methods in classification, we model

$$P(Y = 1|X = x) = 1 - P(Y = 0|X = x) = \Psi(x^T\theta) = \frac{e^{x^T\theta}}{1 + e^{x^T\theta}}, \qquad (1)$$

where $\theta \in \mathbb{R}^p$ is an unknown regression parameter and $\Psi(t) = e^t/(1 + e^t)$ is the logistic function. Suppose we observe $n$ training examples $\{(x_1, y_1), \ldots, (x_n, y_n)\}$.

We study the *sparse high-dimensional* setting, where $n \leq p$ and typically $n \ll p$, and many of the coefficients of $\theta$ are (close to) zero. This setting has been studied by many authors, notably using $\ell_1$-regularized $M$-estimators (e.g. the LASSO), see for instance [3, 22, 30, 31] and the references therein. In Bayesian logistic regression, one assigns a prior distribution to $\theta$, giving a probabilistic model. An especially natural Bayesian way to model sparsity is via a *model selection* prior, which assigns probabilistic weights to every potential model, i.e. every subset of $\{1, \ldots, p\}$ corresponding to selecting the non-zero coordinates of $\theta \in \mathbb{R}^p$. This is a widely used Bayesian approach and includes the hugely popular spike and slab prior [16, 28].

Such priors work well in many settings for estimation and prediction [2, 12, 14], uncertainty quantification [37, 13] and multiple hypothesis testing [11], see [4] for a recent review. An especially attractive property is their interpretability, particularly for variable selection, compared to many other black-box machine learning methods. For example, such methods provide posterior inclusion probabilities of particular features, and their credible sets, which are often important in practice.

However, the discrete nature of such priors makes scalable computation hugely challenging. Under a model selection prior, posterior exploration typically involves searching over all $2^p$ possible models, making standard Markov chain Monte Carlo (MCMC) methods infeasible for moderate $p$ unless the

---

[*]Equal contribution.

feature vectors $\{x_1, \ldots, x_n\}$ satisfy strong structural conditions like orthogonality [14, 46]. There has been recent progress on adapting MCMC methods to sparse high-dimensional logistic regression [29], while another common alternative is to instead use continuous shrinkage-type priors [10, 52].

A popular scalable alternative is variational Bayes (VB), which approximates the posterior by solving an optimization problem. One proposes an approximating family of tractable distributions, called the variational family, and finds the member of this family that is closest to the computationally intractable posterior in Kullback-Leibler (KL) sense. This member is taken as a substitute for the posterior. An especially popular family consists of factorizable distributions, called *mean-field variational Bayes*. VB scales to large data sets and works empirically well in many models, see [7] for a recent review.

In this work, we study the theoretical properties of a mean-field VB approach to sparsity-inducing priors with variational family the set of factorizable spike and slab distributions. This is a natural approximation since it keeps the discrete model selection aspect and many of the interpretable features of the original posterior, while reducing the full $O(2^p)$ model complexity to a much more manageable $O(p)$. The procedure is *adaptive* in that it does not depend on the typically unknown sparsity level, which avoids delicate issues about tuning hyperparameters. This sparse variational family has been employed in various settings [20, 25, 33, 38, 44], including logistic regression [9, 56]. VB is natural in model (1) since in even the simplest low-dimensional setting ($p \ll n$) using Gaussian priors, the posterior is intractable and VB is widely used [6, 21, 34, 43, 49].

However, VB generally comes with few theoretical guarantees, with none currently available in high-dimensional logistic regression. Our main contribution is to show that the sparse VB posterior converges to the true sparse vector at the optimal (minimax) rate in both $\ell_2$ and prediction loss. We prove this under the same conditions for which the true (computationally infeasible) posterior is known to converge [2], showing that one does not necessarily need to sacrifice theoretical guarantees when using sparse VB, at least for estimation and prediction. Our convergence bounds are non-asymptotic and thus reflect relevant finite-sample situations.

Our results also provide practical insights on effective prior and VB calibrations, in particular the choice of prior slab distribution. Many existing works employ Gaussian slabs for the underlying prior, even though these cause excessive shrinkage and suboptimal parameter recovery in benchmark models [14]. Our theoretical results show that optimal parameter recovery is possible if the VB posterior is based on a prior with heavier-tailed Laplace slabs, corroborating findings in linear regression that one should use prior slabs with exponential or heavier tails [12, 14], including for VB [38]. We confirm in simulations that using Laplace prior slabs, as our theory suggests, indeed empirically outperforms the usual VB choice of Gaussian prior slabs, demonstrating the practical importance of the prior slab choice. We further demonstrate that our VB algorithm is empirically competitive with other state-of-the-art Bayesian sparse variable selection methods for logistic regression.

Lastly, we provide conditions on the design matrix under which sparse VB can be expected to work well. Together, these provide theoretical backing for using VB for estimation and prediction in the widely used sparse high-dimensional Bayesian logistic regression model (1).

**Related work**. Theoretical guarantees for VB have been studied for specific models, including linear models [33, 38], exponential family models [47, 48], stochastic block models [55], latent Gaussian models [41] and topic models [17]. In low dimensional settings ($p \ll n$), some Bernstein-von Mises results have also been obtained [26, 50, 51]. In high-dimensional and nonparametric settings, general results have been derived [35, 57] based on the classic Bayesian prior mass and testing approach [18]. There has also been work on studying variational approximations to fractional posteriors [1, 53]. For logistic regression, theoretical results have been established for the fully Bayesian spike and slab approach [2, 29] and its continuous relaxation [52].

Theoretical guarantees for VB in sparse linear regression have recently been obtained in [38]. We combine ideas from this paper with tools from high-dimensional and nonparametric Bayesian statistics [2, 12, 32] to obtain theoretical results in the nonlinear logistic regression model (1). For our algorithm derivation, we use ideas from VB for Bayesian logistic regression [9, 21].

**Organization**. In Section 2 we detail the problem setup, including the notation, prior, variational family and conditions on the design matrix. Main results are found in Section 3, details of the VB algorithm in Section 4, simulations in Section 5 and discussion in Section 6. In the supplement, we present streamlined proofs for the asymptotic results (Section 7), additional simulations (Section 8),

discussion concerning the design matrix conditions (Section 9), full statements and proofs of the non-asymptotic results (Section 10) and a derivation of the VB algorithm (Section 11).

## 2 Problem setup

**Notation**. Recall that we observe $n$ training examples $\{(x_1, y_1), \ldots, (x_n, y_n)\}$ from model (1). For $u \in \mathbb{R}^d$, we write $\|u\|_2 = (\sum_{i=1}^d |u_i|^2)^{1/2}$ for the usual Euclidean norm. Let $X$ be the $n \times p$ matrix with $i^{th}$ row equal to $x_i^T = (x_{i1}, \ldots, x_{ip})$ and for $X_{\cdot j}$ the $j^{th}$ column of $X$, set

$$\|X\| = \max_{1 \leq j \leq p} \|X_{\cdot j}\|_2 = \max_{1 \leq j \leq p} (X^T X)_{jj}^{1/2}.$$

We denote by $P_\theta$ the probability distribution of observing $Y = (Y_1, \ldots, Y_n)$ from model (1) with parameter $\theta \in \mathbb{R}^p$ and by $E_\theta$ the corresponding expectation. For two probability measures $P, Q$, we write $\mathrm{KL}(P||Q) = \int \log \frac{dP}{dQ} dP$ for the Kullback-Leibler divergence. Let $\nabla_\theta f(y, \theta)$ denote the gradient of $f$ with respect to $\theta$. For $\theta \in \mathbb{R}^p$ and a subset $S \subseteq \{1, \ldots, p\}$, we write $|S|$ for the cardinality of $S$ and $\theta_S$ for the vector $(\theta_i : i \in S) \in \mathbb{R}^{|S|}$. We set $S_\theta = \{1 \leq i \leq p : \theta_i \neq 0\}$ to be the set of non-zero coefficients of $\theta$ and write $s_0 = |S_{\theta_0}|$ for the sparsity level of the true parameter $\theta_0$. Throughout the paper we work under the following frequentist assumption:

**Assumption 1.** *There is a true $s_0$-sparse parameter $\theta_0 \in \mathbb{R}^p$ generating the data $Y \sim P_{\theta_0}$.*

### 2.1 Model selection priors and the variational approximation

A model selection prior first selects a dimension $s \in \{0, \ldots, p\}$ from a prior $\pi_p$, then a subset $S \subseteq \{1, \ldots, p\}$ uniformly at random from all $\binom{p}{s}$ subsets of size $|S| = s$, and lastly selects a set of non-zero values for $\theta_S = (\theta_j)_{j \in S} \in \mathbb{R}^{|S|}$ from a product of centered Laplace distributions with density $\prod_{j \in S} \mathrm{Lap}(\lambda)(\theta_j) = (\lambda/2)^{|S|} \exp(-\lambda \sum_{j \in S} |\theta_j|)$, $\lambda > 0$. This prior is represented via the following hierarchical scheme:

$$\begin{aligned}
s &\sim \pi_p(s), \\
S||S| = s &\sim \mathrm{Unif}_{p,s}, \\
\theta_j &\stackrel{ind}{\sim} \begin{cases} \mathrm{Lap}(\lambda), & j \in S, \\ \delta_0, & j \notin S, \end{cases}
\end{aligned} \tag{2}$$

where $\mathrm{Unif}_{p,s}$ selects $S$ from the $\binom{p}{s}$ possible subsets of $\{1, \ldots, p\}$ of size $s$ with equal probability and $\delta_0$ denotes the Dirac mass at zero. We assume that there are constants $A_1, A_2, A_3, A_4 > 0$ with

$$A_1 p^{-A_3} \pi_p(s-1) \leq \pi_p(s) \leq A_2 p^{-A_4} \pi_p(s-1), \qquad s = 1, \ldots, p. \tag{3}$$

Condition (3) is satisfied by a wide range of priors, such as complexity priors [14] and binomial priors, including the widely used spike and slab prior $\theta_j \sim^{iid} \rho \mathrm{Lap}(\lambda) + (1 - \rho)\delta_0$ by taking $\pi_p = \mathrm{Binomial}(p, \rho)$. Assigning a hyperprior $\rho \sim \mathrm{Beta}(1, p^t)$, $t > 1$, to the prior inclusion probabilities also satisfies (3) ([14], Example 2.2), allows mixing over the sparsity level and most importantly gives a spike and slab prior calibration that does not depend on unknown hyperparameters.

While most works use Gaussian slabs for the prior [9, 20, 25, 33, 44, 56], we instead use Laplace slabs since using slab distributions with lighter than exponential tails can cause excessive shrinkage and deteriorate estimation in linear regression [14, 38]. We illustrate numerically that the same phenomenon can occur in logistic regression, where using Laplace rather than Gaussian prior slabs improves estimation, see Section 5. This shows that our theoretical results in Section 3 are reflected in practice and sheds light on suitable prior choices. Full details of the modified algorithm are provided in Algorithm 1 below.

For our theoretical results, we suppose the regularization parameter $\lambda$ of the slab satisfies

$$2\|X\|\sqrt{\log p} \leq \lambda \leq \alpha\|X\|\sqrt{\log p} \tag{4}$$

for some $\alpha \geq 2$. The choice $\lambda \asymp \|X\|\sqrt{\log p}$ is common for the regularization parameter of the LASSO ([8], Chapter 6), which corresponds to the posterior mode based on a full product Laplace prior $\theta_j \sim^{iid} \mathrm{Lap}(\lambda)$ with no extra model selection as in (2). We note additional model selection is

necessary, since the pure Laplace prior can behave badly in sparse high-dimensional settings [14]. Specific values of $\|X\|$ for some design matrices are given in Section 9, but one should typically think of $\|X\| \sim \sqrt{n}$.

Bayesian inference about $\theta_0$, including reconstruction of the class probabilities $P_{\theta_0}(Y = 1 | X = x)$, is carried out via the posterior distribution $\Pi(\cdot|Y)$. Since computing the posterior is infeasible for large $p$, we consider the following mean-field family of approximating distributions

$$\mathcal{Q} = \left\{ Q_{\mu,\sigma,\gamma} = \prod_{j=1}^{p} \left[ \gamma_j N(\mu_j, \sigma_j^2) + (1 - \gamma_j)\delta_0 \right] : \ \gamma_j \in [0,1], \ \mu_j \in \mathbb{R}, \ \sigma_j^2 > 0 \right\}. \quad (5)$$

The VB posterior is the element of $\mathcal{Q}$ that minimizes the KL divergence to the exact posterior

$$Q^* = \arg\min_{Q \in \mathcal{Q}} \mathrm{KL}(Q \| \Pi(\cdot|Y)). \quad (6)$$

The family $\mathcal{Q}$ consists of all factorizable distributions of spike and slab form, which is a natural approximation for sparse settings with variable selection. The $(\gamma_j)$ correspond to the VB variable inclusion probabilities, thereby keeping the interpretability of the original model selection prior. While the prior may factorize like (5), the posterior does not, and we replace the full $2^p$ posterior model weights with the $p$ probabilities $(\gamma_j)$, greatly reducing the posterior dimension. Note that while the *prior* has Laplace slabs, we can fit Gaussian distributions in the variational family since the likelihood induces subgaussian tails in the *posterior*.

Computing the VB posterior $Q^*$ for the variational family $\mathcal{Q}$ in (5) is an optimization problem that has been studied in the literature [20, 25, 33, 38, 44], including for logistic regression [9, 56], mainly using coordinate ascent variational inference (CAVI). While these works mostly consider Gaussian slabs for the prior, CAVI can be suitably modified to the Laplace case, see Algorithm 1.

## 2.2 Design matrix and sparsity assumptions

In the high-dimensional case $p > n$, the parameter $\theta$ in model (1) is not identifiable, let alone estimable, without additional conditions on the design matrix $X$. In the sparse setting, a sufficient condition for consistent estimation is 'local invertibility' of $X^T X$ when restricted to sparse vectors. The following definitions are taken from [2] and make precise this notion of invertibility. Define the diagonal matrix $W \in \mathbb{R}^{n \times n}$ with $i^{th}$ diagonal entry

$$W_{ii} = g''(x_i^T \theta_0) = \Psi(x_i^T \theta_0)(1 - \Psi(x_i^T \theta_0)) \quad (7)$$

and the compatibility type constant

$$\underline{\kappa} = \inf \left\{ \frac{\|W^{1/2} X\theta\|_2^2}{\|X\|^2 \|\theta\|_2^2} : \|\theta_{S_0^c}\|_1 \leq 7\|\theta_{S_0}\|_1, \theta \neq 0 \right\}.$$

For dimension $s \in \{1, \ldots, p\}$, set

$$\overline{\kappa}(s) = \sup \left\{ \frac{\|X\theta\|_2^2}{\|X\|^2 \|\theta\|_2^2} : 0 \neq |S_\theta| \leq s \right\}, \qquad \underline{\kappa}(s) = \inf \left\{ \frac{\|W^{1/2} X\theta\|_2^2}{\|X\|^2 \|\theta\|_2^2} : 0 \neq |S_\theta| \leq s \right\}.$$

For a given $L > 0$ and $\alpha$ defined in (4), we require the following bound on the design matrix

$$\|X\| \geq \alpha \max \left( \frac{50(L+2)\|X\|_\infty}{\underline{\kappa}((L+1)s_0)}, \frac{64}{3\underline{\kappa}} \right) s_0 \sqrt{\log p}. \quad (8)$$

These constants are widely used in the sparsity literature (e.g. [8]), including for high-dimensional logistic regression [2, 31, 52]. Assuming such constants are bounded away from zero and infinity, Atchadé [2] proves that the original posterior $\Pi(\cdot|Y)$ converges to the truth at the optimal rate. We show here that under no further assumptions on the design matrix $X$, the VB posterior $Q^*$ also converges to the truth at the optimal rate. We thus provide theoretical guarantees for the scalable VB approximation under the same conditions for which the true posterior is known to converge.

Many standard design matrices satisfy these compatibility conditions, such as orthogonal designs, i.i.d. (including Gaussian) random matrices and matrices satisfying the 'strong irrepresentability

condition' of [58]. Details of these examples and further discussion are provided in Section 9 in the supplement (see also Chapter 6 of [8]).

For a normalized design matrix with entries of size $O(1)$, one has $\|X\| \sim \sqrt{n}$, so that (8) is a minimal sample size condition. For suitably bounded compatibility constants, this translates into the minimal sample size $n \gtrsim s_0^2 \log p$, as in [2]. The frequentist $\ell_1$-regularized $M$-estimator is known to converge at the same rate under similar assumptions to ours for a deterministic design matrix $X$ [22] and under slightly weaker sample size conditions for an i.i.d. subgaussian random design matrix $X$ ($n \gtrsim s_0 \log p$) [31].

## 3 Main results

We now provide theoretical guarantees for the VB posterior $Q^*$ in (6). We present here our results in a simpler asymptotic form as $n, p \to \infty$ for easier readability. More complicated, but practically more relevant, finite sample guarantees are provided in Section 10 of the supplement. In particular, one should keep in mind that the results here do indeed reflect finite-sample behaviour.

We investigate how well the VB posterior recovers the true underlying high-dimensional parameter $\theta_0$. This is measured via the speed of posterior concentration, which studies the size of the *smallest* $\ell_2$ or prediction type neighbourhood around the true $\theta_0$ that contains most of the (VB) posterior probability [18]. This is a frequentist assessment that describes the typical behaviour of the VB posterior under the true generative model, see Assumption 1.

Posterior concentration rates are now entering the machine learning community as tools to gain insights into (variational) Bayesian methods and assess the suitability of priors and their calibrations (e.g. [35, 36, 42, 51]). Such results also quantify the typical distance between a point estimator $\hat{\theta}$ (posterior mean/median) and the truth ([18], Theorem 2.5), as well as the typical posterior spread about the truth. Taken together, these quantities are crucial for the accuracy of Bayesian uncertainty quantification and so good posterior concentration results are necessary conditions for ensuring the latter. Ideally, most of the posterior probability should be concentrated in a ball centered around the true $\theta_0$ with radius proportional to the optimal (minimax) rate. This is the case for the true computationally infeasible posterior [2] and extends to the VB posterior $Q^*$, as we now show. This provides a universal, objective guarantee for the VB posterior.

**Theorem 1.** *Suppose the model selection prior* (2) *satisfies* (3) *and* (4) *and the design matrix $X$ satisfies assumption* (8) *for some sequence $L = M_n$. Then the VB posterior $Q^*$ satisfies*

$$E_{\theta_0} Q^* \left( \theta \in \mathbb{R}^p : \|\theta - \theta_0\|_2 \geq \frac{M_n^{1/2}}{\underline{\kappa}(M_n s_0)} \frac{\sqrt{s_0 \log p}}{\|X\|} \right) = O\big(C_\kappa/M_n\big) + o(1),$$

*where $C_\kappa = L_0\big(\frac{\bar{\kappa}(L_0 s_0)}{\underline{\kappa}((1+4L_0/A_4)s_0)^2} + \underline{\kappa}(L_0 s_0)^{-1}\big)$ and $L_0 = 2\max\{A_4/5, (1.1 + 4\alpha^2/\underline{\kappa} + 2A_4 + \log(4 + \bar{\kappa}(s_0)))/A_4\}$. Define the mean-squared prediction error $\|p_\theta - p_0\|_n^2 = \frac{1}{n}\sum_{i=1}^n (\Psi(x_i^T \theta) - \Psi(x_i^T \theta_0))^2$, where we recall $P_\theta(Y = 1|X = x) = \Psi(x_i^T \theta)$. Then the VB posterior $Q^*$ satisfies*

$$E_{\theta_0} Q^* \left( \theta \in \mathbb{R}^p : \|p_\theta - p_0\|_n \geq \frac{\sqrt{M_n \bar{\kappa}(M_n s_0)}}{\underline{\kappa}(M_n s_0)} \sqrt{\frac{s_0 \log p}{n}} \right) = O\big(C_\kappa/M_n\big) + o(1).$$

*In particular, if $C_\kappa/M_n \to 0$, then the posterior concentrates around the true sparse parameter $\theta_0$ at the optimal (minimax) rate in both $\ell_2$ and mean-squared prediction loss.*

**Remark.** *In Theorem 1, we keep track of the compatibility numbers in the rate. If $\bar{\kappa}(L_0 s_0)$, $\underline{\kappa}((1 + 4L_0/A_4)s_0)$ and $\underline{\kappa}(L_0 s_0)$ are bounded away from zero and infinity, as is often the case, the right hand side of both displays tends to zero for any $M_n \to \infty$ growing arbitrary slowly. In this case, the rates simplify to $M_n^{1/2}\sqrt{s_0 \log p}/\|X\|$ and $M_n\sqrt{s_0 \log p/n}$, respectively. In Section 9 of the supplement, we give sufficient conditions for this to happen.*

Since $\sqrt{s_0 \log p}/\|X\|$ is the minimax rate, this result says that for estimating $\theta_0$, in either $\ell_2$ or prediction loss, the VB approximation behaves optimally from a theoretical frequentist perspective. In fact, since these conditions are essentially the same as were used to study the true posterior [2], our results suggest one does not lose much by using this computationally more efficient approximation, at

least regarding estimation and prediction. This backs up the empirical evidence that VB can provide excellent scalable estimation. For a non-asymptotic version of Theorem 1, see Theorem 4 in Section 10.

Since the prior and variational family do not depend on unknown parameters (e.g. sparsity level $s_0 = |S_{\theta_0}|$ of $\theta_0$), the procedure is *adaptive*, i.e. it can recover an $s_0$-sparse truth nearly as well as if we knew the exact true sparsity level beforehand. This avoids difficult issues about tuning parameter selection. As mentioned above, posterior concentration results can be used to obtain guarantees for point estimators, such as the VB posterior mean.

**Theorem 2.** *Assume the conditions of Theorem 1 and that $n \geq 1.1 + 4\alpha^2/\underline{\kappa} + 2A_4 + \log(4 + \overline{\kappa}(s_0)))$. Then the VB predictive mean $\hat{p}^*(x) = \int P_\theta(Y = 1|X = x)dQ^*(\theta) = \int \Psi(x^T\theta)dQ^*(\theta)$ and true prediction function $p_0(x) = P_{\theta_0}(Y = 1|X = x)$ satisfy*

$$P_{\theta_0}\left(\|\hat{p}^* - p_0\|_n \geq \frac{M_n^{1/2}\overline{\kappa}(\frac{2n}{A_4 \log p} + s_0)^{1/2}}{\underline{\kappa}(\frac{2n}{A_4 \log p} + (1 - 2/A_4)s_0)}\sqrt{\frac{s_0 \log p}{n}}\right) = O(C_\kappa/M_n) + o(1).$$

**Remark.** *If $\|X\| \sim \sqrt{n}$, as is often the case, then the extra sample size condition in Theorem 2 is automatically implied by* (8), *see Section 2.2. We have again kept track of the compatibility numbers; if these are bounded away from zero and infinity, then the probability converges to zero and we recover the rate $M_n^{1/2}\sqrt{s_0 \log p/n}$ for any $M_n \to \infty$ arbitrarily slowly.*

We next show the VB posterior $Q^*$ does not provide overly conservative model selection in the sense that it concentrates on models of size at most a constant multiple of the true model size $s_0$. This gives some guarantees for variable selection, in particular bounding the number of false positives. It provides a first theoretical underpinning for interpretable inference when using this VB approximation.

**Theorem 3.** *Under the conditions of Theorem 1, the VB posterior satisfies*

$$E_{\theta_0}Q^*(\theta \in \mathbb{R}^p : |S_\theta| \geq M_n s_0) = O(C_\kappa/M_n) + o(1).$$

We have thus shown the VB posterior $Q^*$ (1) concentrates at the optimal rate around the sparse truth in both $\ell_2$ and prediction loss and (2) does not select overly large models, with the VB posterior mean sharing property (1). These provide some reassuring theoretical guarantees regarding the behaviour of this scalable and interpretable sparse VB approximation. Our results are reflected in practice, as our VB algorithm performs better empirically than commonly used sparse VB approaches, see Section 5.

## 4 Variational algorithm

We present a coordinate-ascent variational inference (CAVI) algorithm to compute the VB posterior $Q^*$ in (6). Consider the prior (2) with $\theta_j \sim^{iid} (1-w)\delta_0 + w\text{Lap}(\lambda)$ and hyperprior $w \sim \text{Beta}(a_0, b_0)$. Introducing binary latent variables $(z_j)_{j=1}^p$, this spike and slab prior has hierarchical representation

$$\begin{aligned}
w &\sim \text{Beta}(a_0, b_0), \\
z_j|w &\sim^{iid} \text{Bernoulli}(w), \\
\theta_j|z_j &\sim^{ind} (1 - z_j)\delta_0 + z_j\text{Lap}(\lambda).
\end{aligned} \tag{9}$$

Minimizing the objective (6) is intractable for Bayesian logistic regression, so we instead minimize a surrogate objective obtained by maximizing a lower bound on the marginal likelihood following the ideas of [21, 9], see Section 11 for details. This common approach is known to lead to improved accuracy in approximation [21], see also Chapter 10.6 of [6]. The surrogate objective is non-convex, as is typically the case in VB, so the CAVI algorithm can be sensitive to initialization [7] and parameter updating order [38]. Introducing a free parameter $\eta \in \mathbb{R}^n$, we can establish the upper bound

$$\text{KL}(Q_{\mu,\sigma,\gamma}||\Pi(\cdot|Y)) \leq \text{KL}(Q_{\mu,\sigma,\gamma}||\Pi) - E_{\theta \sim Q_{\mu,\sigma,\gamma}}[f(\theta, \eta)] \tag{10}$$

for a suitable function $f$ defined in (30). We minimize the right-hand side over the variational family $Q_{\mu,\sigma,\gamma} \in \mathcal{Q}$, i.e. over the parameters $\mu, \sigma, \gamma$. Since we seek the tightest possible upper bound in (10), we also minimize this over the free parameter $\eta$. In particular, CAVI alternates between updating

$\eta$ for fixed $\mu, \sigma, \gamma$ and then cycling through $\mu_j, \sigma_j, \gamma_j$ and updating these while keeping all other parameters fixed. Keeping all other parameters fixed, one updates $\mu_j$ and $\sigma_j$ by minimizing

$$
\begin{aligned}
\mu_j \mapsto \quad & \lambda\sigma_j\sqrt{\frac{2}{\pi}}e^{-\frac{\mu_j^2}{2\sigma_j^2}} + \lambda\mu_j\mathrm{erf}\left(\frac{\mu_j}{\sqrt{2}\sigma_j}\right) + \mu_j^2\sum_{i=1}^{n}\frac{1}{4\eta_i}\tanh(\eta_i/2)x_{ij}^2 \\
& + \mu_j\left(\sum_{i=1}^{n}\frac{1}{2\eta_i}\tanh(\eta_i/2)x_{ij}\sum_{k\neq j}\gamma_k x_{ik}\mu_k - \sum_{i=1}^{n}(y_i - 1/2)x_{ij}\right), \quad (11)
\end{aligned}
$$

$$
\sigma_j \mapsto \quad \lambda\sigma_j\sqrt{\frac{2}{\pi}}e^{-\frac{\mu_j^2}{2\sigma_j^2}} + \lambda\mu_j\mathrm{erf}\left(\frac{\mu_j}{\sqrt{2}\sigma_j}\right) - \log\sigma_j + \sigma_j^2\sum_{i=1}^{n}\frac{1}{4\eta_i}\tanh(\eta_i/2)x_{ij}^2,
$$

respectively, where $\mathrm{erf}(x) = 2/\sqrt{\pi}\int_0^x e^{-t^2}\,dt$ is the error function. One updates $\gamma_j$ by solving

$$
\begin{aligned}
-\log\frac{\gamma_j}{1-\gamma_j} = \; & \log\frac{b_0}{a_0} + \lambda\sigma_j\sqrt{\frac{2}{\pi}}e^{-\frac{\mu_j^2}{2\sigma_j^2}} + \lambda\mu_j\mathrm{erf}\left(\frac{\mu_j}{\sqrt{2}\sigma_j}\right) - \mu_j\sum_{i=1}^{n}(y_i - 1/2)x_{ij} - \frac{1}{2} \\
& + \sum_{i=1}^{n}\frac{1}{4\eta_i}\tanh(\eta_i/2)\left(x_{ij}^2(\mu_j^2 + \sigma_j^2) + 2x_{ij}\mu_j\sum_{k\neq j}\gamma_k x_{ik}\mu_k\right) - \log(\lambda\sigma_j) \quad (12)
\end{aligned}
$$

and updates $\eta$ via

$$
\eta_i^2 = E_{\mu,\gamma,\sigma}(x_i^T\theta)^2 = \sum_{k=1}^{p}\gamma_k x_{ik}^2(\mu_k^2 + \sigma_k^2) + \sum_{k=1}^{p}\sum_{l\neq k}(\gamma_k x_{ik}\mu_k)(\gamma_l x_{il}\mu_l). \quad (13)
$$

A full derivation of (10)-(13) can be found in Section 11 In our implementation, we minimize (11) using the limited memory Broyden-Fletcher-Goldfarb-Shanno (BFGS) algorithm [24]. To improve scalability, we also perform the parameter updates in parallel, updating the objective function parameter values only at the end of each iteration. Details are provided in Algorithm 1.

---

**Algorithm 1:** `Modified CAVI for variational Bayes with Laplace slabs`

---

**Input:** $X \in \mathbb{R}^{n\times p}, Y \in \{0,1\}^p$
$(\mu, \sigma, \gamma) \leftarrow \texttt{init\_param}()$
**while** !convergence **do**
    $\eta \leftarrow \texttt{update\_lower\_bound}(\mu, \sigma, \gamma)$ ;                   `// implements eq.(13)`
    $\texttt{obj\_fun} \leftarrow \texttt{generate\_obj\_fun}(\mu, \sigma, \gamma, \eta, X, Y)$
    **for** $j \in [p]$ **do**
        $(\mu_j, \sigma_j) \leftarrow \texttt{L\_BFGS}\big(\texttt{obj\_fun.at}(j)\big)$ ;         `// minimizes eq.(11)`
    **end**
    $\gamma \leftarrow \texttt{update\_alpha}(\mu, \sigma, \gamma, \eta, X, Y)$ ;              `// implements eq.(12)`
**end**
**Output:** $\mu \in \mathbb{R}^p, \sigma^2 \in \mathbb{R}_{>0}^p, \gamma \in [0,1]^p$

---

## 5 Numerical study

We empirically compare the performance of our VB method **VB (Lap)** based on prior (9) with parameters $a_0 = b_0 = \lambda = 1$, to other state-of-the-art Bayesian variable selection methods in a simple simulation study. We implemented Algorithm 1 in C++ using the Rcpp interface and used the Armadillo linear algebra library and ensmallen optimization library, see [5, 39]. Note that the VB objective function is highly non-convex and so the local minimum returned by Algorithm 1 does not necessarily equal the global minimizer to the VB optimzation problem (6). For comparison, we first consider the usual VB approach **VB (Gauss)**, where the same variational family (5) is used, but the Laplace slabs are replaced by standard normal distributions in the prior (9) (e.g. [20, 25, 33, 44, 56]). We also compare our approach with the **varbvs** [9], **SkinnyGibbs** [29], **BinaryEMVS** [27], **BhGLM** [54] and **rstanarm** [19] packages. We provide a brief description of these methods in Section 8.1 in the supplement. Note that the choice of hyperparameters can affect each of these methods and

performance gains are possible from using good data-driven choices. For instance, our algorithm is sensitive to the choice of $\lambda$, see Section 8.3 in the supplement.

Due to space constraints, we provide here only one test case and defer the remaining tests to Section 8.2 of the supplement. We take $n = 250$, $p = 500$ and $X$ to be a standard Gaussian design matrix, i.e. $X_{ij} \sim^{iid} N(0,1)$, and set the true signal $\theta_0 = (2, 2, 0, \ldots, 0)^T$ to be $s = 2$ sparse. We ran the experiment 200 times for each method and report the means and standard deviations of the following performance measures: (i) true positive rate (TPR), (ii) false discovery rate (FDR), (iii) $\ell_2$-loss of the posterior mean $\hat{\theta}$, i.e. $\|\hat{\theta} - \theta_0\|_2$, (iv) mean-squared predictive error (MSPE) of the posterior mean, i.e. $(\frac{1}{n} \sum_{i=1}^{n} |\Psi(x_i^T \hat{\theta}) - \Psi(x_i^T \theta_0)|^2)^{1/2}$ and (v) run time in seconds. Since the BhGLM and rstanarm packages do not have explicit model selection subroutines, the TPR and FDR are not applicable. Similarly, SkinnyGibbs provides posterior model selection probabilities and not the posterior mean, hence neither the $\ell_2$-error or MSPE are applicable. The results are in Table 1.

Table 1: Comparing sparse Bayesian methods in high-dimensional logistic regression.

| Algorithm | TPR | FDR | $\ell_2$-error | MSPE | Time |
|---|---|---|---|---|---|
| VB (Lap) | **1.00 ± 0.00** | **0.03 ± 0.10** | **0.57 ± 0.37** | **0.04 ± 0.02** | 12.10 ± 0.49 |
| VB (Gauss) | **1.00 ± 0.00** | 0.84 ± 0.04 | 3.34 ± 0.46 | 0.25 ± 0.03 | 0.98 ± 0.51 |
| varbvs | **1.00 ± 0.00** | 0.92 ± 0.01 | 1.30 ± 0.15 | 0.16 ± 0.02 | **0.08 ± 0.01** |
| SkinnyGibbs | **1.00 ± 0.00** | 0.90 ± 0.01 | – | – | 22.58 ± 3.58 |
| BinEMVS | **1.00 ± 0.00** | 0.88 ± 0.03 | 4.25 ± 0.56 | 0.29 ± 0.02 | 31.04 ± 0.94 |
| BhGLM | – | – | 2.69 ± 0.88 | 0.21 ± 0.16 | 1.60 ± 0.37 |
| rstanarm | – | – | 0.74 ± 0.58 | 0.31 ± 0.24 | 197.20 ± 26.16 |

Results based on 200 runs for an i.i.d. standard Gaussian design matrix $X \in \mathbb{R}^{250 \times 500}$ with $X_{ij} \sim^{iid} N(0,1)$ and true signal $\theta_{0,1} = \theta_{0,2} = 2$, $\theta_{0,j} = 0$ for $3 \le j \le 500$.

In Table 1 and the additional simulations in Section 8, we see that using Laplace slabs in the prior (9) generally outperforms the commonly used Gaussian slabs in all statistical metrics ($\ell_2$-loss, MPSE, FDR), in some cases substantially so. This highlights the empirical advantages of using Laplace rather than Gaussian slabs for the prior underlying the VB approximation and matches the theory presented in Section 3, as well as similar observations in linear regression [38]. We also highlight the excellent FDR of VB (Lap), which warrants further investigation given its importance in Bayesian variable selection and multiple hypothesis testing.

However, the computational run-time is substantially slower for Laplace slabs due to the absence of analytically tractable update formulas as in the Gaussian case. The optimization routines required in Algorithm 1 mean a naive implementation can significantly increase the run-time; we are currently working on a more efficient implementation as an R-package `sparsevb` [15] that should reduce the run-time by at least an order of magnitude.

The other methods perform roughly similarly to our algorithm in terms of estimation ($\ell_2$-error) and prediction (MSPE) error, doing better in certain test cases and worse in others. It seems there is no clearly dominant Bayesian approach regarding accuracy. However, our method provides the best results concerning model selection, generally having the best FDR while maintaining a competitive TPR, as suggested by Theorem 3. The other methods all perform comparably, with varbvs having the best TPR but substantially higher FDR, meaning it identifies many coefficients to be significant, both correctly and incorrectly.

We note that the two VB methods based on Gaussian prior slabs (VB (Gauss) and varbvs), which use analytic update formulas, are all significantly faster than the other methods. All the VB methods (including ours) scale much better with larger model sizes (e.g. $p = 5000$) than the other methods, which did not finish running in a reasonable amount of time when $p = 5000$, see Section 8.2. As expected, the MCMC methods generally performed slowest, though SkinnyGibbs, which is designed to scale up MCMC to larger sparse models, is indeed an order of magnitude faster than rstanarm.

## 5.1 Coverage of marginal credible intervals

An advantage of Bayesian methods is their ability to perform uncertainty quantification via credible sets. The present mean-field VB approximation provides access to marginal credible sets for individual features, which can be more informative than just the VB posterior mean, and are often of interest to

practitioners. However, VB is known to generally underestimate the posterior variance, which can lead to bad uncertainty quantification. In view of the excellent FDR control of our VB method in earlier simulations, we further investigate the performance of these marginal credible sets empirically.

We consider 4 tests cases, consisting of the above example (Test 0) and Tests 1-3 from Section 8.2. In each case, we computed 95% marginal credible intervals for the coefficients, i.e. the intervals $I_j$, $j = 1, \ldots, p$, of smallest length such that $Q^*(\theta_j \in I_j) \geq 0.95$. We ran each experiments 200 times and report the mean and standard deviation of the coverage and length of these credible intervals for both the (true) zero and non-zero coefficients in Table 2.

Table 2: Marginal VB credible intervals for individual features

|  | Test 0 | Test 1 | Test 2 | Test 3 |
|---|---|---|---|---|
| Coverage (non-zero coefficients) | $1.00 \pm 0.00$ | $0.82 \pm 0.19$ | $0.94 \pm 0.11$ | $0.38 \pm 0.12$ |
| Length (non-zero coefficients) | $2.87 \pm 0.14$ | $2.56 \pm 0.20$ | $2.27 \pm 0.16$ | $1.33 \pm 0.32$ |
| Coverage (zero coefficients) | $1.00 \pm 0.00$ | $1.00 \pm 0.00$ | $1.00 \pm 0.00$ | $0.99 \pm 0.01$ |
| Length (zero coefficients) | $0.00 \pm 0.00$ | $0.01 \pm 0.01$ | $0.00 \pm 0.00$ | $0.03 \pm 0.01$ |

$X \in \mathbb{R}^{250 \times 500}$, $X_{ij} \sim^{iid} N(0, \sigma^2)$. (0) $\sigma = 1$, $s = 2$, $\theta_{0,1:s} = 2$; (1) $\sigma = 0.25$, $s = 5$, $\theta_{0,1:s} = 4$; (2) $\sigma = 2$, $s = 10$, $\theta_{0,1:s} = 6$; (3) $\sigma = 0.5$, $s = 15$, $\theta_{0,1:s} \sim^{iid}$ Unif$(-2, 2)$.

We see that for the true zero coefficients $\theta_{0,j} = 0$, the coverage is close to one with intervals of nearly zero width, meaning the VB posterior sets $\gamma_j = Q^*(\theta_j \neq 0) < 0.05$. This matches the other evidence (Theorem 3 and the good FDR) that the VB posterior $Q^*$ does not include too many spurious variables. For true non-zero coefficients $\theta_{0,j} \neq 0$, the coverage is moderate to excellent in the first 3 experiments, which matches the good TPR seen above. However, coverage is low in Test 3, which is an especially difficult test typically containing several small non-zero coefficients (here $\theta_{0,j} \sim^{iid}$ Unif$(-2, 2)$) that are hard to detect. The low coverage is not surprising, since it is known that even in the full Bayes case, coefficients below a certain size cannot be consistently covered [45].

The very promising results here suggest our VB approach might be effective for uncertainty quantification for individual features, however this requires further investigation. We lastly note that while it may be reasonable to consider marginal credible intervals, one should be careful about using more general VB credible sets due to the mean-field approximation.

## 6   Discussion

This paper investigates a scalable and interpretable mean-field variational approximation of the popular spike and slab prior with Laplace slabs in high-dimensional logistic regression. We derive theoretical guarantees for this approach, proving (1) optimal concentration rates for the VB posterior in $\ell_2$ and prediction loss around a sparse true parameter, (2) optimal convergence rates for the VB posterior predictive mean and (3) that the VB posterior does not select overly large models, thereby controlling the number of false discoveries.

We verify in a numerical study that the empirical performance of the proposed method reflects these theoretical guarantees. In particular, using Laplace slabs in the prior underlying the variational approximation can substantially outperform the same VB method with Gaussian prior slabs, as is typically used in the literature, though at the expense of slower computation. The proposed approach performs comparably with other state-of-the-art sparse high-dimensional Bayesian variable selection methods for logistic regression, but scales substantially better to high-dimensional models where other approaches based on the EM algorithm or MCMC are not computable. We are currently working on a more efficient implementation as an R-package `sparsevb` [15] that should improve the run-time.

Based on the promising FDR control and coverage of our VB method in the simulations, we plan to further investigate the theoretical and empirical performance of our algorithm for multiple hypothesis testing and variable selection, see [11] for promising first results for the (unscalable) original posterior. Furthermore, the results derived here are the first steps towards better understanding VB methods in sparse high-dimensional nonlinear models. It opens up several interesting future lines of research for applying scalable VB implementations of spike and slab priors in complex high-dimensional models, including Bayesian neural networks [36], (causal) graphical models [23] and high-dimensional Bayesian time series [40].

**Acknowledgements**: We thank 4 reviewers for their useful comments that helped improve the presentation of this work. Botond Szabó received funding from the Netherlands Organization for Scientific Research (NWO) under Project number: 639.031.654.

**Impact statement**: Our theoretical results seek to better understand how sparse VB approximations work and thus improve their performance and reliability in practice. Since our results have no specific applications in mind, seeking rather to explain and improve an existing method, any potential broader impact will derive from improved performance in fields where such methods are already used.

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
