[Supplementary Material]

# Supplementary material for 'Spike and slab variational Bayes for high dimensional logistic regression'

**Kolyan Ray**[*]
Department of Mathematics
Imperial College London
kolyan.ray@ic.ac.uk

**Botond Szabó**[*]
Department of Mathematics
Vrije Universiteit Amsterdam
b.t.szabo@vu.nl

**Gabriel Clara**
Department of Mathematics
Vrije Universiteit Amsterdam
g.clara@student.vu.nl

In this supplement we present streamlined proofs for the asymptotic results (Section 7), additional simulations (Section 8), discussion concerning the design matrix conditions (Section 9), full statements and proofs of the non-asymptotic results (Section 10) and a derivation of the VB algorithm (Section 11).

## 7 Proofs

Our proofs use the following general result, which allows us to bound the VB probabilities of sets having exponentially small probability under the true posterior.

**Lemma 1** (Theorem 5 of [16]). *Let $\Theta_n$ be a subset of the parameter space, $A$ be an event and $Q$ be a distribution for $\theta$. If there exists $C > 0$ and $\delta_n > 0$ such that*

$$E_{\theta_0}\Pi(\theta \in \Theta_n|Y)1_A \leq Ce^{-\delta_n}, \tag{14}$$

*then*

$$E_{\theta_0}Q(\theta \in \Theta_n)1_A \leq \tfrac{2}{\delta_n}\left[E_{\theta_0}KL(Q||\Pi(\cdot|Y))1_A + Ce^{-\delta_n/2}\right].$$

We must thus show that there exist events $(A_n)$ satisfying $P_{\theta_0}(A_n) \to 1$ such that on $A_n$:

1. the posterior puts at most exponentially small probability $Ce^{-\delta_n}$ outside $\Theta_n$,
2. the KL divergence between the VB posterior $Q^*$ and true posterior is $O(\delta_n)$.

To aid readability, in this section we state all intermediate results in asymptotic form, keeping track of only the leading order terms as $n, p \to \infty$. We provide full non-asymptotic statements in Section 10, which may be skipped on first reading, though most of the technical difficulty is contained in these results.

For $t, L, M_1, M_2 > 0$, define the events

$$\mathcal{A}_{n,1}(t) = \{\|\nabla_\theta \ell_{n,\theta_0}(Y)\|_\infty \leq t\},$$
$$\mathcal{A}_{n,2}(L) = \{\Pi(\theta \in \mathbb{R}^p : |S_\theta| \leq Ls_0|Y) \geq 3/4\},$$
$$\mathcal{A}_{n,3}(M_1, M_2) = \{\Pi(\theta \in \mathbb{R}^p : \|\theta - \theta_0\|_2 > M_1\sqrt{s_0 \log p}/\|X\|\,|Y) \leq e^{-M_2 s_0 \log p}\}.$$

The first event bounds the score function $\nabla_\theta \ell_{n,\theta_0}(Y)$ and is needed to control the first order term in the Taylor expansion of the log-likelihood, see (20). The second says the posterior concentrates on models of size at most a constant multiple of the true model dimension. The last event says the posterior places all but exponentially small probability on an $\ell_2$-ball of the optimal radius about the truth and is used for a localization argument.

---

[*]Equal contribution.

The event required to apply Lemma 1 is defined by

$$\mathcal{A}_n(t, L, M_1, M_2) = \mathcal{A}_{n,1}(t) \cap \mathcal{A}_{n,2}(L) \cap \mathcal{A}_{n,3}(M_1, M_2). \tag{15}$$

The proof has an iterative structure, where we localize the posterior based on the event $\mathcal{A}_{n,i-1}$ to prove $\mathcal{A}_{n,i}$ has high probability. The idea to iteratively localize the posterior during the proofs is a useful technique from Bayesian nonparametrics (e.g. [14]).

**Lemma 2.** *Suppose the prior satisfies* (3) *and* (4), *and the design matrix satisfies condition* (8) *for* $L = 2\max\{A_4/5, (1.1 + 4\alpha^2/\underline{\kappa} + 2A_4 + \log(4 + \overline{\kappa}(s_0)))/A_4\}$. *Then for $p$ large enough,*

$$P_{\theta_0}\Big(\mathcal{A}_n(t, L, M_1, M_2)^c\Big) = O(1/p),$$

*with* $t = \|X\|\sqrt{\log p}$, $M_1 = D_0\sqrt{L}/\underline{\kappa}((1+4L/A_4)s_0)$, $D_0 = (24/\sqrt{A_4})\max(25\alpha, \frac{1+A_3}{16}, \frac{3A_4+4}{32})$, *and* $M_2 = 2L$.

The next two lemmas establish exponential posterior bounds on the event $\mathcal{A}_{n,1}(\|X\|\sqrt{\log p}) \supset \mathcal{A}_n(t, L, M_1, M_2)$. The first states that the posterior concentrates on models of size at most a constant multiple of the true model size $s_0 = |S_{\theta_0}|$. The second states that the posterior concentrates on an $\ell_2$-ball of optimal radius about the true parameter $\theta_0$.

**Lemma 3.** *Suppose the prior satisfies* (3) *and* (4). *If for* $L \geq 2(1.1 + 4\alpha^2/\underline{\kappa} + 2A_4 + \log(4 + \overline{\kappa}(s_0)))/A_4$ *the design matrix satisfies condition* (8), *then for $p$ large enough,*

$$E_{\theta_0}[\Pi\left(\theta \in \mathbb{R}^p : |S_\theta| \geq Ls_0 \mid Y\right) 1_{\mathcal{A}_{n,1}(\|X\|\sqrt{\log p})}] \leq 2\exp\left\{-(LA_4/2)s_0\log p\right\}.$$

**Lemma 4.** *Suppose the prior satisfies* (3) *and* (4). *If for* $K > \max\{A_4, 2(1.1 + 4\alpha^2/\underline{\kappa} + 2A_4 + \log(4 + \overline{\kappa}(s_0))/A_4\}$, *the design matrix satisfies* (8) *with* $L = 2K/A_4$, *then for $p$ large enough,*

$$E_{\theta_0}\left[\Pi\left(\theta \in \mathbb{R}^p : \|\theta - \theta_0\|_2 \geq \frac{D_1\sqrt{K}}{\underline{\kappa}\big((2K/A_4 + 1)s_0\big)}\frac{\sqrt{s_0\log p}}{\|X\|}\Big| Y\right) 1_{\mathcal{A}_{n,1}(\|X\|\sqrt{\log p})}\right] \leq 8e^{-Ks_0\log p},$$

*where* $D_1 = 16A_4^{-1/2}\max\{25\alpha, \frac{3+2A_2+A_3}{16}\}$.

Lemmas 2, 3 and 4 follow immediately from their non-asymptotic counterparts Lemmas 11, 8 and 10, respectively, in Section 10. We finally control the KL divergence between the VB posterior $Q^*$ and posterior $\Pi(\cdot|Y)$ on the event $\mathcal{A}_n$; see Lemma 7 for the corresponding non-asymptotic result. This is the most difficult technical step in establishing our result.

**Lemma 5.** *Consider the event* $\mathcal{A}_n = \mathcal{A}_n(t, L, M_1, M_2)$ *in Lemma 2. Then for sufficiently large $p$, the VB posterior $Q^*$ satisfies*

$$KL(Q^*||\Pi(\cdot|Y))1_{\mathcal{A}_n} \leq D_2s_0\log p,$$

*with* $D_2 = L(\frac{(9D_0/4+\alpha/2)D_0\bar{\kappa}(Ls_0)}{\underline{\kappa}((1+4L/A_4)s_0)^2} + \frac{\alpha}{2\underline{\kappa}(Ls_0)})$ *and where* $L, D_0$ *are given in Lemma 2.*

We briefly explain the heuristic idea behind the proof of Lemma 5. Since the VB posterior $Q^*$ is the minimizer of the KL objective (6), $KL(Q^*||\Pi(\cdot|Y)) \leq KL(Q||\Pi(\cdot|Y))$ for any $Q \in \mathcal{Q}$ in the variational family. We upper bound this quantity for some $Q$ carefully chosen according to the logistic likelihood. We first identify a model $S \subseteq \{1, \ldots, p\}$ which is not too far from the true model $S_{\theta_0}$ and to which the posterior assigns sufficient, though potentially exponentially small, probability. In the low dimensional setting $p \ll n$, Taylor expanding the log-likelihood $\ell_{n,\theta} - \ell_{n,\theta_0}$ asymptotically gives a Gaussian linear regression likelihood with rescaled design matrix and data. This motivates the distribution $Q \in \mathcal{Q}$ which fits a normal distribution $N_S(\mu_S, D_S)$ on $S$ with mean $\mu_S$ equal to the least squares estimator solving the linearized Gaussian approximation and covariance $D_S$ (a diagonalized version of) the covariance matrix of this estimator, again in the linearized model. While the Taylor expansion is not actually valid in the sparse high-dimensional setting $p \gg n$ considered here, we can nonetheless show that the approximation is still sufficiently good to apply Lemma 1.

*Proof of Theorem 1.* We apply Lemma 1 with

$$\Theta_n = \left\{\theta \in \mathbb{R}^p : \|\theta - \theta_0\|_2 \geq \frac{M_n^{1/2}\sqrt{s_0\log p}}{\underline{\kappa}(M_ns_0)\|X\|}\right\}$$

and $A = \mathcal{A}_n = \mathcal{A}_n(t, L, M_1, M_2)$ the event in Lemma 2. Since $\mathcal{A}_n \subset \mathcal{A}_{n,1}(\|X\|\sqrt{\log p})$, Lemma 4 implies that $E_{\theta_0}\Pi(\theta \in \Theta_n | Y)1_{\mathcal{A}_n} \leq 8e^{-D_1^{-2}M_n s_0 \log p}$, i.e. (14) holds with $\delta_n = D_1^{-2}M_n s_0 \log p$. Using Lemma 1 followed by Lemma 5, since $\delta_n \to \infty$,

$$E_{\theta_0}Q^*(\theta \in \Theta_n)1_{\mathcal{A}_n} \leq \tfrac{2}{\delta_n}\left[E_{\theta_0}\mathrm{KL}(Q^*\|\Pi(\cdot|Y))1_{\mathcal{A}_n} + 8e^{-\delta_n/2}\right] \leq \frac{2\alpha^2 D_2}{M_n}(1 + o(1)) = O(C_\kappa/M_n).$$

Since $P_{\theta_0}(\mathcal{A}_n) \to 1$ by Lemma 2, the first result follows. Since $\|\Psi'\|_\infty \leq 1/4$,

$$n\|p_\theta - p_0\|_n^2 \leq \tfrac{1}{16}\sum_{i=1}^n |x_i^T(\theta - \theta_0)|^2 = \tfrac{1}{16}\|X(\theta - \theta_0)\|_2^2 \leq \tfrac{1}{16}\overline{\kappa}((M_n+1)s_0)\|X\|^2\|\theta - \theta_0\|_2^2, \quad (16)$$

where the last inequality follows from Theorem 3 and the definition of $\overline{\kappa}(\cdot)$. The second statement then follows by combining the first statement of the theorem with the above display. $\square$

*Proof of Theorem 2.* By the duality formula for the KL divergence ([3], Corollary 4.15),

$$\int f(\theta)dQ^*(\theta) \leq \mathrm{KL}(Q^*\|\Pi(\cdot|Y)) + \log\int e^{f(\theta)}d\Pi(\theta|Y)$$

for any measurable $f$ such that $\int e^{f(\theta)}d\Pi(\theta|Y) < \infty$. Let $\mathcal{A}_n = \mathcal{A}(t, L, M_1, M_2)$ be the event in Lemma 2. Applying Jensen's inequality and taking $f(\theta) = cn\|p_\theta - p_0\|_n^2$ for $c > 0$,

$$cn\|\hat{p}^*- p_0\|_n^2 1_{\mathcal{A}_n} = cn\|E^{Q^*}p_\theta - p_0\|_n^2 1_{\mathcal{A}_n} \leq E^{Q^*}cn\|p_\theta - p_0\|_n^2 1_{\mathcal{A}_n}$$

$$\leq \mathrm{KL}(Q^*\|\Pi(\cdot|Y))1_{\mathcal{A}_n} + 1_{\mathcal{A}_n}\log\int e^{cn\|p_\theta - p_0\|_n^2}d\Pi(\theta|Y). \qquad (17)$$

By Lemma 5, the first term is bounded by $D_2 s_0 \log p$. For notational convenience, write $\overline{\kappa}^* = \overline{\kappa}(\frac{2n}{A_4 \log p} + s_0)$ and $\underline{\kappa}^* = \underline{\kappa}(\frac{2n}{A_4 \log p} + (1 - 2/A_4)s_0)$. For $D_1$ defined in Lemma 4 and $K > 0$ the minimal constant satisfying the conditions of Lemma 4, set

$$\mathcal{B}_0 = \left\{\theta \in \mathbb{R}^p : |S_\theta| \leq \tfrac{2n}{A_4 \log p} - 1, n\|p_\theta - p_0\|_n^2 \leq K\frac{D_1^2 \overline{\kappa}^* s_0 \log p}{(\underline{\kappa}^*)^2}\right\},$$

$$\mathcal{B}_j = \left\{\theta \in \mathbb{R}^p : |S_\theta| \leq \tfrac{2n}{A_4 \log p} - 1, j\frac{D_1^2 \overline{\kappa}^* s_0 \log p}{(\underline{\kappa}^*)^2} < n\|p_\theta - p_0\|_n^2 \leq (j+1)\frac{D_1^2 \overline{\kappa}^* s_0 \log p}{(\underline{\kappa}^*)^2}\right\},$$

$$\bar{\mathcal{B}} = \left\{\theta \in \mathbb{R}^p : |S_\theta| > \tfrac{2n}{A_4 \log p} - 1\right\}.$$

Since $E[U1_\Omega] = E[U|\Omega]P(\Omega)$, conditional Jensen's inequality gives

$$E[(\log V)1_\Omega] \leq P(\Omega)\log E[V|\Omega] = P(\Omega)\log E[V1_\Omega] - P(\Omega)\log P(\Omega)$$

for any random variable $V$ and event $\Omega$. The $E_{\theta_0}$-expectation of the second term in (17) thus equals

$$E_{\theta_0}\left[\log\left(\int_{\mathcal{B}_0}e^{cn\|p_\theta - p_0\|_n^2}d\Pi(\theta|Y) + \sum_{j=K}^{n/(s_0\log p)-1}\int_{\mathcal{B}_j}e^{cn\|p_\theta - p_0\|_n^2}d\Pi(\theta|Y)\right.\right.$$

$$\left.\left. + \int_{\bar{\mathcal{B}}}e^{cn\|p_\theta - p_0\|_n^2}d\Pi(\theta|Y)\right)1_{\mathcal{A}_n}\right]$$

$$\leq P_{\theta_0}(\mathcal{A}_n)\log E_{\theta_0}\left[e^{cK\frac{D_1^2 \overline{\kappa}^* s_0 \log p}{(\underline{\kappa}^*)^2}} + \sum_{j=K}^{n/(s_0\log p)-1}e^{c(j+1)\frac{D_1^2 \overline{\kappa}^* s_0 \log p}{(\underline{\kappa}^*)^2}}\Pi(\mathcal{B}_j|Y)1_{\mathcal{A}_n} + e^{cn}\Pi(\bar{\mathcal{B}}|Y)1_{\mathcal{A}_n}\right]$$

$$ - P_{\theta_0}(\mathcal{A}_n)\log P_{\theta_0}(\mathcal{A}_n).$$

Using (16) and Lemma 4 gives $E_{\theta_0}[\Pi(\mathcal{B}_j|Y)1_{\mathcal{A}_n}] \leq 8e^{-js_0\log p}$, while Lemma 3 gives $E_{\theta_0}[\Pi(\bar{\mathcal{B}}|Y)1_{\mathcal{A}_n}] \leq 2e^{-n}$. Taking $c = c_n = \frac{(\underline{\kappa}^*)^2}{2D_1^2 \overline{\kappa}^*} \leq 1/(128)$ and using that $P_{\theta_0}(\mathcal{A}_n^c) = O(1/p)$ by Lemma 2, the last display is bounded by,

$$\log\left[e^{(K/2)s_0\log p} + 8\sum_{j=K}^{n/(s_0\log p)-1}e^{(\frac{j+1}{2}-j)s_0\log p} + 2e^{-(1-c)n}\right] + O(1/p) \leq Ks_0\log p(1 + o(1)).$$

Plugging in the preceding bounds for the right hand side of (17),

$$nE_{\theta_0}\|\hat{p}^* - p_0\|_n^2 1_{\mathcal{A}_n} \leq c_n^{-1}(K + D_2)s_0 \log p(1 + o(1)) = O(c_n^{-1}C_\kappa s_0 \log p),$$

for $C_\kappa$ the constant in Theorem 1. Applying Markov's inequality and Lemma 2,

$$P_{\theta_0}(\|\hat{p}^* - p_0\|_n \geq r) \leq r^{-2}E_{\theta_0}[\|\hat{p}^* - p_0\|_n^2 1_{\mathcal{A}_n}] + P_{\theta_0}(\mathcal{A}_n^c) = O(r^{-2}n^{-1}c_n^{-1}C_\kappa s_0 \log p) + o(1).$$

Taking $r = M_n^{1/2}(\overline{\kappa}^*)^{1/2}(\underline{\kappa}^*)^{-1}\sqrt{s_0 \log p/n}$ gives the result. $\qquad\square$

*Proof of Theorem 3.* This follows the same argument as the proof of Theorem 1, taking $\Theta_n = \{\theta \in \mathbb{R}^p : |S_\theta| \geq M_n s_0\}$, $\delta_n = (A_4/2)M_n s_0 \log p$ and applying Lemma 3 instead of Lemma 4. $\qquad\square$

# 8 Additional numerical results

## 8.1 Description of Bayesian methods for logistic regression

In the **varbvs** package [5], the standard VB algorithm was implemented for the prior (2) with Gaussian instead of Laplace slabs and an additional layer of importance sampling for computing the low-dimensional prior hyperparameters. As for our algorithm, we set $a_0 = b_0 = 1$. In the **BinaryEMVS** package [11], an expectation-maximization (EM) algorithm is implemented for fitting Bayesian spike and slab regularization paths for logistic regression. More concretely, the considered spike and slab prior takes the form

$$\omega \sim \text{Beta}(a_0, b_0),$$
$$z_j \sim^{iid} \text{Bern}(\omega),$$
$$\theta_j \sim^{iid} (1 - z_j)\mathcal{N}(0, \sigma_1^2) + z_j\mathcal{N}(0, \sigma_2^2), \qquad \text{with } \sigma_1 \ll \sigma_2.$$

In the simulation study, we use the parametrization $a_0 = 1$, $b_0 = 1$, $\sigma_1 = 0.025$ and $\sigma_2 = 5$. In the Bayesian hierarchical generalized linear model (**BhGLM** package) of [18], an EM algorithm is implemented for a mixture of Laplace priors:

$$\omega \sim \text{Unif}[0, 1],$$
$$z_j \sim^{iid} \text{Bern}(\omega),$$
$$\theta_j \sim^{iid} (1 - z_j)\text{Lap}(\lambda_1) + z_j\text{Lap}(\lambda_2), \qquad \text{with } \lambda_1 \gg \lambda_2.$$

We also work with the default parametrization of the bmlasso function of the BhGLM package, i.e. $s_1 = 1/\lambda_1 = 0.04$ and $s_2 = 1/\lambda_2 = 0.5$. We use the implementation of **SkinnyGibbs** in the supplementary material to [12], using the function skinnybasad with the settings provided in the example from the manual, apart from taking pr=0.5 to reflect the present prior setting $a_0 = b_0 = 1$. Finally, we consider the **rstanarm** package, which makes Bayesian regression modeling via the probabilistic programming language Stan accessible in R. In the package, Hamiltonian Monte Carlo [8] was implemented for the horseshoe global-local shrinkage prior [6]. We run the function stan_glm, again with the default parameterization, i.e. we set the global scale to 0.01 with degree of freedom 1, and the slab-scale to 2.5 with degrees of freedom 4. The default inferential algorithm runs 4 Markov chains with 2000 iterations each.

## 8.2 Additional experiments

We provide five further test cases in addition to the experiment considered in Section 5. In all cases we consider Gaussian design matrices, but vary all other parameter. In tests 1-3, we take $n = 250$ and $p = 500$ as in Section 5, while in experiments 4 and 5 we set $n = 2500$ and $p = 5000$. The entries of the design matrices have independent centered Gaussian distributions with standard deviations $\sigma = 0.25$, 2, 0.5, 0.5, 1, respectively. The true underlying signal has sparsity levels $s = 5$, 10, 15, 25, 25, respectively, with the non-zero signal coefficients located at the beginning of the signal with values equal to (1) $\theta_{0,j} = 4$, (2) $\theta_{0,j} = 6$, (3) $\theta_{0,j} \sim^{iid} \text{Unif}(-2, 2)$, (4) $\theta_{0,j} = 2$ and (5) $\theta_{0,j} \sim^{iid} \text{Unif}(-1, 1)$ for $j = 1, \ldots, s$.

We ran each experiment 200 times and report the means and standard deviations of the performance metrics in Table 3. The $\ell_2$-error ($\ell_2(\hat{\theta}, \theta_0) = \|\hat{\theta} - \theta_0\|_2$) and mean squared prediction

error ($\mathrm{MSPE}(\hat{p}) = (\frac{1}{n}\sum_{i=1}^{n} |\Psi(x_i^T\hat{\theta}) - \Psi(x_i^T\theta_0)|^2)^{1/2}$) are reported with respect to the posterior mean. For the methods performing model selection, we use the standard threshold 0.5 for the marginal posterior inclusion probability $\alpha_j$, i.e. the posterior includes the $j$th coefficient in the model if $\alpha_j > 0.5$. The true positive rate (TPR) and false discovery rate (FDR) are then defined as $\mathrm{TPR}(\alpha) = s^{-1}\sum_{j:\theta_{0,j}\neq 0} 1_{\alpha_j > 0.5}$ and $\mathrm{FDR}(\alpha) = \sum_{j:\alpha_j > 0.5} 1_{\theta_{0,j}=0}/|\{j : \alpha_j > 0.5\}|$, respectively. The elapsed times are given for an Intel i7-8550u laptop processor.

As in Section 5, we conclude that our VB approach typically outperforms the other variational algorithms using prior Gaussian slabs, though again at the expense of greater computational times. Compared to our approach, the other four methods based on the EM algorithm or MCMC performed better in some scenarios and worse in others, but with substantially greater computational times. Unsurprisingly, the rstanarm package using MCMC is the slowest; in many cases it did not even converge after 8000 iterations. In the high-dimensional case $p = 5000$ and $n = 2500$, 4 algorithms (SkinnyGibbs, BhGLM, BinEMVS, rstanarm) did not finish the computations in a reasonable amount of time: the fastest required at least 100 hours to execute 200 runs and for rstanarm, even a single run required multiple hours.

Table 3: Comparing sparse Bayesian methods in high-dimensional logistic regression.

| | Algorithm | Test 1 | Test 2 | Test 3 | Test 4 | Test 5 |
|---|---|---|---|---|---|---|
| TPR | VB (Lap) | $0.99 \pm 0.06$ | $\mathbf{1.00 \pm 0.00}$ | $0.51 \pm 0.11$ | $\mathbf{1.00 \pm 0.00}$ | $0.40 \pm 0.28$ |
| | VB (Gauss) | $1.00 \pm 0.01$ | $1.00 \pm 0.02$ | $0.54 \pm 0.11$ | $\mathbf{1.00 \pm 0.00}$ | $0.85 \pm 0.06$ |
| | varbvs | $\mathbf{1.00 \pm 0.00}$ | $\mathbf{1.00 \pm 0.00}$ | $\mathbf{0.68 \pm 0.11}$ | $\mathbf{1.00 \pm 0.00}$ | $\mathbf{0.87 \pm 0.06}$ |
| | SkinnyGibbs | $0.98 \pm 0.06$ | $1.00 \pm 0.02$ | $0.51 \pm 0.12$ | – | – |
| | BinEMVS | $0.99 \pm 0.03$ | $\mathbf{1.00 \pm 0.00}$ | $0.58 \pm 0.11$ | – | – |
| FDR | VB (Lap) | $0.49 \pm 0.11$ | $\mathbf{0.00 \pm 0.02}$ | $\mathbf{0.41 \pm 0.14}$ | $\mathbf{0.01 \pm 0.02}$ | $\mathbf{0.03 \pm 0.05}$ |
| | VB (Gauss) | $0.63 \pm 0.07$ | $0.09 \pm 0.13$ | $0.52 \pm 0.12$ | $0.81 \pm 0.02$ | $0.95 \pm 0.01$ |
| | varbvs | $0.93 \pm 0.01$ | $0.08 \pm 0.08$ | $0.83 \pm 0.03$ | $0.93 \pm 0.00$ | $0.91 \pm 0.01$ |
| | SkinnyGibbs | $0.80 \pm 0.03$ | $0.11 \pm 0.11$ | $0.71 \pm 0.07$ | – | – |
| | BinEMVS | $\mathbf{0.43 \pm 0.14}$ | $0.19 \pm 0.10$ | $0.63 \pm 0.10$ | – | – |
| $\ell_2$-Error | VB (Lap) | $3.97 \pm 0.85$ | $\mathbf{1.73 \pm 0.59}$ | $4.89 \pm 1.29$ | $\mathbf{4.23 \pm 0.72}$ | $2.31 \pm 0.64$ |
| | VB (Gauss) | $4.99 \pm 0.46$ | $13.82 \pm 0.16$ | $3.86 \pm 0.61$ | $8.34 \pm 0.49$ | $17.05 \pm 0.52$ |
| | varbvs | $7.29 \pm 0.24$ | $16.31 \pm 0.06$ | $3.35 \pm 0.44$ | $7.68 \pm 0.03$ | $\mathbf{1.32 \pm 0.13}$ |
| | BhGLM | $4.39 \pm 0.65$ | $15.68 \pm 0.53$ | $\mathbf{2.81 \pm 0.46}$ | – | – |
| | BinEMVS | $3.84 \pm 0.97$ | $14.49 \pm 0.28$ | $5.82 \pm 0.89$ | – | – |
| | rstanarm | $\mathbf{2.87 \pm 1.49}$ | $6.74 \pm 0.86$ | $3.14 \pm 0.88$ | – | – |
| MSPE | VB (Lap) | $0.18 \pm 0.02$ | $0.07 \pm 0.02$ | $0.24 \pm 0.03$ | $\mathbf{0.06 \pm 0.01}$ | $0.22 \pm 0.09$ |
| | VB (Gauss) | $0.21 \pm 0.02$ | $0.08 \pm 0.02$ | $0.25 \pm 0.03$ | $0.21 \pm 0.01$ | $0.34 \pm 0.01$ |
| | varbvs | $0.21 \pm 0.01$ | $0.12 \pm 0.01$ | $\mathbf{0.19 \pm 0.02}$ | $0.18 \pm 0.00$ | $\mathbf{0.17 \pm 0.01}$ |
| | BhGLM | $0.22 \pm 0.15$ | $\mathbf{0.06 \pm 0.05}$ | $0.20 \pm 0.16$ | – | – |
| | BinEMVS | $\mathbf{0.15 \pm 0.03}$ | $0.09 \pm 0.02$ | $0.27 \pm 0.03$ | – | – |
| | rstanarm | $0.36 \pm 0.25$ | $0.10 \pm 0.08$ | $0.37 \pm 0.30$ | – | – |
| Time | VB (Lap) | $8.66 \pm 6.19$ | $12.05 \pm 0.55$ | $14.53 \pm 0.78$ | $360.22 \pm 10.97$ | $358.79 \pm 8.43$ |
| | VB (Gauss) | $0.23 \pm 0.05$ | $1.45 \pm 0.66$ | $1.70 \pm 1.37$ | $356.76 \pm 7.22$ | $359.78 \pm 1.02$ |
| | varbvs | $\mathbf{0.05 \pm 0.00}$ | $\mathbf{0.10 \pm 0.03}$ | $\mathbf{0.03 \pm 0.01}$ | $\mathbf{4.29 \pm 0.12}$ | $\mathbf{8.15 \pm 0.48}$ |
| | SkinnyGibbs | $19.89 \pm 0.04$ | $19.77 \pm 0.09$ | $20.30 \pm 0.42$ | – | – |
| | BhGLM | $1.37 \pm 0.28$ | $2.24 \pm 0.71$ | $1.42 \pm 0.52$ | – | – |
| | BinEMVS | $12.43 \pm 4.28$ | $36.52 \pm 4.82$ | $30.44 \pm 2.88$ | – | – |
| | rstanarm | $152.46 \pm 18.00$ | $426.20 \pm 50.67$ | $181.53 \pm 26.05$ | – | – |

The design matrices $X \in \mathbb{R}^{n\times p}$ are taken to be $X_{ij} \sim^{iid} N(0, \sigma^2)$. The signal vector $\theta_0$ has $s$ non-zero coefficients, all located at the beginning of the signal.
(1) $X \in \mathbb{R}^{250\times 500}$, $\sigma = 0.25$, $s = 5$, $\theta_{0,1:s} = 4$
(2) $X \in \mathbb{R}^{250\times 500}$, $\sigma = 2$, $s = 10$, $\theta_{0,1:s} = 6$
(3) $X \in \mathbb{R}^{250\times 500}$, $\sigma = 0.5$, $s = 15$, $\theta_{0,1:s} \sim^{iid} \mathrm{Unif}(-2, 2)$
(4) $X \in \mathbb{R}^{2500\times 5000}$, $\sigma = 0.5$, $s = 25$, $\theta_{0,1:s} = 2$
(5) $X \in \mathbb{R}^{2500\times 5000}$, $\sigma = 1$, $s = 10$, $\theta_{0,1:s} \sim^{iid} \mathrm{Unif}(-1, 1)$

## 8.3 Comparing different choices of the hyperparameter $\lambda$

Theorems 1-3 state that for the hyperparameter choice $\lambda \asymp \|X\|\sqrt{\log p}$, our VB algorithm has good asymptotic properties. In practice, however, the finite-sample performance indeed depends on $\lambda$. We ran our algorithm 200 times on the experiment considered in Section 5 for different choices of $\lambda$ and report the results in Table 4. In this example, the performance was sensitive to the choice of $\lambda$ with large values of $\lambda$, which cause more shrinkage, performing worse.

In linear regression, where more extensive simulations have been carried out, the choice of $\lambda$ was similarly found to have an effect, though there was not clear evidence to support a particular fixed choice of $\lambda$, with larger values sometimes performing better and sometime worse [16]. This suggests using a data-driven choice of $\lambda$ may be helpful in practice, for example using cross validation.

Table 4: Varying the scale hyperparameter

|  | $\lambda = \frac{1}{20}$ | $\lambda = \frac{1}{5}$ | $\lambda = 2$ | $\lambda = 5$ | $\lambda = 20$ |
|---|---|---|---|---|---|
| TPR | $1.00 \pm 0.00$ | $1.00 \pm 0.00$ | $1.00 \pm 0.00$ | $1.00 \pm 0.00$ | $0.81 \pm 0.28$ |
| FDR | $0.00 \pm 0.00$ | $0.02 \pm 0.08$ | $0.03 \pm 0.10$ | $0.09 \pm 0.16$ | $0.02 \pm 0.10$ |
| $\ell_2$-error | $0.53 \pm 0.36$ | $0.58 \pm 0.37$ | $0.48 \pm 0.28$ | $0.39 \pm 0.14$ | $1.73 \pm 0.44$ |
| MSPE | $0.04 \pm 0.02$ | $0.04 \pm 0.02$ | $0.04 \pm 0.02$ | $0.04 \pm 0.01$ | $0.17 \pm 0.07$ |
| Time | $12.13 \pm 0.58$ | $12.02 \pm 0.73$ | $11.93 \pm 0.71$ | $11.72 \pm 1.05$ | $12.10 \pm 0.73$ |

$X \in \mathbb{R}^{250 \times 500}$, $X_{ij} \sim^{iid} N(0,1)$, $s = 2$, $\theta_{0,1:s} = 2$

## 9 Further discussion of the design matrix and sparsity assumptions

For $s \in \{1, \ldots, p\}$, recall the definitions

$$\underline{\kappa} = \inf \left\{ \frac{\|W^{1/2}X\theta\|_2^2}{\|X\|^2\|\theta\|_2^2} : \|\theta_{S_0^c}\|_1 \leq 7\|\theta_{S_0}\|_1, \theta \neq 0 \right\},$$

$$\overline{\kappa}(s) = \sup \left\{ \frac{\|X\theta\|_2^2}{\|X\|^2\|\theta\|_2^2} : 0 \neq |S_\theta| \leq s \right\}, \qquad \underline{\kappa}(s) = \inf \left\{ \frac{\|W^{1/2}X\theta\|_2^2}{\|X\|^2\|\theta\|_2^2} : 0 \neq |S_\theta| \leq s \right\}.$$

These definitions are taken from the sparsity literature [4] and are used in sparse logistic regression [1, 13, 17]. We reproduce some of the discussion here for convenience, but refer the reader to Chapter 6 of [4] for further reading.

The true model $S_0$ is *compatible* if $\underline{\kappa} > 0$, which implies $\|W^{1/2}X\theta\|_2^2 \geq \underline{\kappa}\|X\|^2\|\theta\|_2^2$ for all $\theta$ in the relevant set. The number 7 can be altered and is taken to match the conditions used in [1, 7] since we use some of their results. Compatibility $\underline{\kappa} > 0$ involves approximate rather than exact sparsity, since the parameters $\theta$ need only have small rather than zero coordinates outside $S_0$. In contrast, $\underline{\kappa}(s)$ involves exactly $s$-sparse vectors. Note that if $\|X\| = 1$, then $\underline{\kappa}(s)^{1/2}$ equals the smallest scaled singular value of a submatrix of $W^{1/2}X$ of dimension $s$. Similarly, $\overline{\kappa}(s)^{1/2}$ upper bounds the operator norm of $X$ when restricted to exactly $s$-sparse vectors.

Even though $W$ depends on the unknown $\theta_0$, it does not necessarily play a significant role in the above definitions. If $\|X\theta_0\|_\infty$ is bounded, then the true regression function $P_{\theta_0}(Y = 1|X = x_i) = \Psi(X_i^T\theta_0)$ is bounded away from zero and one at the design points and $W$ is equivalent to the identity matrix $I_n$. One can then set $W = I_n$ in the above definitions by simply rescaling the constants. Note that estimation in classification problems is known to behave qualitatively differently near the boundary points 0 and 1, see, e.g. [15].

When $W = I_n$, we recover the exact compatibility constants used in sparse linear regression [7, 16]. This is natural since when linearizing the logistic regression model, the likelihood asymptotically looks like that of a linear regression model with design matrix $W^{1/2}X$, see Section 7. One therefore expects similar conditions with $X$ replaced by $W^{1/2}X$. For further discussion, see Chapter 6 of [4] or Section 2.2 of [7]; in particular, Lemma 1 of [7] provides a concise relation between various notions of compatibility.

Another common condition in the sparsity literature is the *mutual coherence* of $X$, which equals the largest correlation between its columns:

$$\mathrm{mc}(X) = \max_{1 \leq i \neq j \leq p} \frac{|\langle X_{\cdot i}, X_{\cdot j} \rangle|}{\|X_{\cdot i}\|_2 \|X_{\cdot j}\|_2} = \max_{1 \leq i \neq j \leq p} \frac{|(X^T X)_{ij}|}{\|X_{\cdot i}\|_2 \|X_{\cdot j}\|_2}.$$

Conditions of this nature have been used by many authors (see Section 2.2 of [7] for references) and measure how far from orthogonal the matrix $X$ is. One can relate the present compatibility constants to the mutual coherence.

**Lemma 6.** *Suppose $\|X\theta_0\|_\infty \leq R$ is bounded and $\min_{1 \leq i \neq j \leq p} \frac{\|X_{\cdot i}\|_2}{\|X_{\cdot j}\|_2} \geq \eta$. Then for $C = C(R)$,*

$$\overline{\kappa}(s) \leq 1 + smc(X), \qquad \underline{\kappa} \geq C(R)(\eta^2 - 64 s_0 mc(X)), \qquad \underline{\kappa}(s) \geq C(R)(\eta^2 - smc(X)).$$

*Proof.* For $\theta$ an $s$-sparse vector, using Cauchy-Schwarz,

$$\|X\theta\|_2^2 = \sum_{j=1}^p \left( (X^T X)_{jj} \theta_j^2 + \sum_{k \neq j} \theta_j (X^T X)_{jk} \theta_k \right) \leq \|X\|^2 \|\theta\|_2^2 + \mathrm{mc}(X)\|X\|^2 \|\theta\|_1^2$$

$$\leq (1 + smc(X))\|X\|^2\|\theta\|_2^2,$$

so that $\overline{\kappa}(s) \leq 1 + smc(X)$. For $R > 0$, one has $\Psi(-R) \geq e^{-R}/2$ and $\Psi(R) \leq 1 - e^{-R}/2$, so that $\Psi(x_i^T \theta_0) \in [e^{-R}/2, 1 - e^{-R}/2]$. Using the definition (7), all diagonal entries of $W$ satisfy $W_{ii} \in [\delta_R, 1/4]$ for $\delta_R = e^{-R}(2 - e^{-R})/4 > 0$, so that $\|W^{1/2} X\theta\|_2^2 \geq \delta_R \|X\theta\|_2^2$. It thus suffices to prove the result with $W = I_n$ at the expense of the factor $\delta_R$. With $W = I_n$, arguing as in Lemma 1 of [7] gives $\underline{\kappa} \geq \eta^2 - 64 s_0 \mathrm{mc}(X)$ and $\underline{\kappa}(s) \geq \eta^2 - smc(X)$. $\square$

If $\|X\theta_0\|_\infty$ is bounded and

$$s_0 = o(1/\mathrm{mc}(X)), \qquad \min_{1 \leq i \neq j \leq p} \frac{\|X_{\cdot i}\|_2}{\|X_{\cdot j}\|_2} \geq \eta > 0, \qquad (18)$$

namely the truth is sufficiently sparse and the column norms of $X$ are comparable, then $\overline{\kappa}(Ls_0) \leq 1 + o(1)$, $\underline{\kappa} \gtrsim \eta^2 - o(1)$ and $\underline{\kappa}(Ls_0) \gtrsim \eta^2 - o(1)$ for any $L > 0$, as required for the results in this paper. Condition (18) has been considered in [16], and thus the various examples in [16] are also covered by our results, including:

- (Orthogonal design). If $X$ is an orthogonal matrix with $\langle X_{\cdot i}, X_{\cdot j} \rangle = 0$ for $i \neq j$ with suitably normalized column lengths $\|X_{\cdot i}\|_2 = \sqrt{n}$.
- (IID responses). Suppose the original matrix entries are i.i.d. random variables $W_{ij}$ and set $X_{ij} = \sqrt{n} W_{ij}/\|W_{\cdot j}\|_2$, so that the columns are normalized to have length $\sqrt{n}$. If $|W_{ij}| \leq C$ almost surely and $\log p = o(n)$, then (18) holds for sparsity levels $s_0 = o(\sqrt{n/\log p})$. Similarly, if $E e^{t|W_{ij}|^\gamma} < \infty$ for some $\gamma, t > 0$ and $\log p = o(n^{\gamma/(4+\gamma)})$, then (18) again holds for $s_0 = o(\sqrt{n/\log p})$. This covers the standard Gaussian random design $W_{ij} \sim^{iid} N(0,1)$ if $\log p = o(n^{1/3})$. See [16] for details.
- Rescale the columns as in the IID response model so that $\|X_{\cdot i}\|_2 = \sqrt{n}$ for all $i$. Then the $p \times p$ matrix $C = X^T X/n$ takes values one on its diagonal and $C_{ij}$, $i \neq j$, equals the correlation between columns $i$ and $j$. If either $C_{ij} = r$ for a constant $0 < r < (1 + cm)^{-1}$ and all $i \neq j$, or $|C_{ij}| \leq \frac{c}{2m-1}$ for every $i \neq j$, then $\mathrm{mc}(X) = \max_{i \neq j} C_{ij} = O(1/m)$ and so (18) holds for sparsity level $s_0 = o(m)$. Such matrices are studied in Zhao and Yu [19], who show that models up to dimension $m$ satisfy the 'strong irrepresentability condition'.

For further details of why these satisfy (18), and hence our conditions, see Section 2.2 of [16].

## 10 Non-asymptotic results and proofs

This section contains the non-asymptotic formulations of all the technical results used in this paper, together with their proofs. These results imply the more digestible asymptotic formulations presented

in Section 3. To begin, recall the log-likelihood in model (1) based on data $Y = (Y_1, \dots, Y_n)$ is

$$\ell_{n,\theta}(Y) = \sum_{i=1}^{n} Y_i x_i^T \theta - g(x_i^T \theta) = \sum_{i=1}^{n} Y_i (X\theta)_i - g((X\theta)_i), \qquad (19)$$

where $g(t) = \log(1 + e^t)$, $t \in \mathbb{R}$. We use the following notation for the first-order remainder of the Taylor expansion of the log-likelihood:

$$\mathcal{L}_{n,\theta}(y) := \ell_{n,\theta}(y) - \ell_{n,\theta_0}(y) - \nabla_\theta \ell_{n,\theta_0}(y)^T (\theta - \theta_0). \qquad (20)$$

## 10.1 The Kullback-Leibler divergence between $Q^*$ and $\Pi(\cdot|Y)$

To apply Lemma 1, we must bound $\text{KL}(Q^*||\Pi(\cdot|Y))$ on the event $\mathcal{A}_n$ given in (15). We do this by bounding the KL divergence between the posterior and a carefully selected element of the variational family. We choose a spike and slab distribution whose slab is centered at the least squares estimator of the linearized logistic likelihood approximation and whose covariance equals the (diagonalized) covariance of this estimator. This builds on ideas in [16], extending them to the nonlinear logistic regression model.

For a given model $S \subseteq \{1, \dots, p\}$, we write $X_S$ for the $n \times |S|$-submatrix of $X$ keeping only the columns $X_{\cdot i}$, $i \in S$, and $\theta_S \in \mathbb{R}^{|S|}$ for the vector $(\theta_i : i \in S)$.

**Lemma 7.** *Consider the event $\mathcal{A}_n = \mathcal{A}_n(t, L, M_1, M_2)$ in (15) with $M_2 > L$. If $(4e)^{1/(M_2 - L)} \leq p^{s_0}$, then the variational Bayes posterior $Q^*$ satisfies*

$$KL(Q^*||\Pi(\cdot|Y))1_{\mathcal{A}_n} \leq \zeta_n,$$

*where*

$$\zeta_n = s_0 \log p \left( L + \frac{9}{4} M_1^2 \overline{\kappa}(Ls_0) + \frac{\lambda L^{1/2}}{\|X\|\sqrt{\log p}} \left( 2M_1 + \frac{tL^{1/2}}{4\underline{\kappa}(Ls_0)\|X\|\sqrt{\log p}} + \frac{L^{1/2}}{\sqrt{\underline{\kappa}(1)\log p}} \right) \right)$$
$$+ Ls_0 \log \frac{1}{4\underline{\kappa}(Ls_0)} + \log(2e).$$

*Proof.* Since the VB posterior $Q^*$ minimizes the KL objective (6), we have $\text{KL}(Q^*||\Pi(\cdot|Y)) \leq \text{KL}(Q||\Pi(\cdot|Y))$ for all $Q \in \mathcal{Q}$. It thus suffices to bound this last KL divergence for a suitably chosen element $Q \in \mathcal{Q}$, which may (and will) depend on the true unknown parameter $\theta_0$, on the event $\mathcal{A}_n = \mathcal{A}_n(t, L, M_1, M_2)$.

Recall that the posterior distribution is a mixture over all possible submodels and can thus be written

$$\Pi(\cdot|Y) = \sum_{S \subseteq \{1, \dots, p\}} \hat{w}_S \Pi_S(\cdot|Y) \otimes \delta_{S^c}, \qquad (21)$$

where the posterior model weights satisfy $0 \leq \hat{w}_S \leq 1$ and $\sum_S \hat{w}_S = 1$ and $\Pi_S(\cdot|Y)$ denotes the posterior for $\theta_S \in \mathbb{R}^{|S|}$ in the restricted model with $Y_i \sim \text{Bin}(1, \Psi((X_S \theta_S)_i))$, i.e. the logistic regression model (1) with $\theta_{S^c} = 0$.

**Choosing $Q' \in \mathcal{Q}$.** Recall that $E_{\theta_0} Y_i = \Psi(x_i^T \theta_0)$ and for a model $S \subseteq \{1, \dots, p\}$ set

$$\mu_S = \theta_{0,S} + (X_S^T X_S)^{-1} X_S^T (Y - E_{\theta_0} Y), \qquad \Sigma_S = (X_S^T X_S)^{-1}, \qquad (22)$$

and define the $|S| \times |S|$ diagonal matrix $D_S$ by

$$(D_S)_{jj} = \frac{1}{(\Sigma_S^{-1})_{jj}} = \frac{1}{(X_S^T X_S)_{jj}}, \qquad (23)$$

$j = 1, \dots, |S|$. We choose as element of our variational family the distribution

$$Q'(\theta) = N_{S'}(\mu_{S'}, D_{S'})(\theta_{S'}) \times \delta_{S'^c}(\theta_{S'^c}) = \prod_{j \in S'} N\left(\mu_{S',j}, \frac{1}{(X_{S'}^T X_{S'})_{jj}}\right)(\theta_i) \prod_{j \in S'^c} \delta_0(\theta_i),$$

for a model $S' \subseteq \{1, \dots, p\}$ satisfying the following three properties:

$$|S'| \leq Ls_0, \qquad \|\theta_{0,S'^c}\|_2 \leq M_1 \sqrt{s_0 \log p}/\|X\|, \qquad \hat{w}_{S'} \geq (2e)^{-1} p^{-Ls_0}, \qquad (24)$$

where $L, M_1$ are the constants in $\mathcal{A}_n(t, L, M_1, M_2)$. Note that $Q'$ is indeed an element of the mean-field variational family $\mathcal{Q}$ in (5) with $\gamma_j = 1$ if $i \in S'$ and $\gamma_j = 0$ otherwise.

**Existence of $S'$ satisfying** (24). We first show that on the event $\mathcal{A}_n$, there exists a subset $S'$ satisfying (24), so that our choice of $Q'$ is indeed valid. On $\mathcal{A}_n$,

$$\Pi(\theta : \|\theta_{0,S_\theta^c}\|_2 > M_1\sqrt{s_0 \log p}/\|X\| \mid Y) \leq \Pi(\theta : \|\theta - \theta_0\|_2 > M_1\sqrt{s_0 \log p}/\|X\| \mid Y)$$
$$\leq e^{-M_2 s_0 \log p} \to 0,$$

so that the posterior model weights satisfy

$$\sum_{\substack{S:|S|\leq Ls_0 \\ \|\theta_{0,S^c}\|_2 \leq M_1\sqrt{s_0 \log p}/\|X\|}} \hat{w}_S \geq 3/4 - e^{-M_2 s_0 \log p} \geq 1/2$$

on $\mathcal{A}_n$, since $e^{-M_2 s_0 \log p} \leq (4e)^{-M_2/(M_2-L)} \leq (4e)^{-1} \leq 1/4$ by assumption. Using $\binom{p}{s} \leq p^s/s!$, the number of elements in the last sum is bounded by

$$\sum_{S:|S|\leq Ls_0} 1 \leq \sum_{s=0}^{Ls_0} \binom{p}{s} \leq \sum_{s=0}^{Ls_0} \frac{p^s}{s!} \leq ep^{Ls_0},$$

which implies that on $\mathcal{A}_n$ there exists a set $S' \subseteq \{1, \ldots, p\}$ of size $|S'| \leq Ls_0$ with $\|\theta_{0,S'^c}\|_2 \leq M_1\sqrt{s_0 \log p}/\|X\|$ and with posterior probability $\hat{w}_{S'} \geq (2e)^{-1}p^{-Ls_0}$, i.e. satisfying (24).

**Reduction to the non-diagonal covariance case.** Since $Q'$ is only absolutely continuous with respect to the $\hat{w}_{S'}\Pi_{S'}(\cdot|Y) \otimes \delta_{S'^c}$ term of the posterior (21),

$$\mathrm{KL}(Q'||\Pi(\cdot|Y)) = E_{\theta \sim N_{S'}(\mu_{S'}, D_{S'}) \otimes \delta_{S'^c}} \log \frac{dN_{S'}(\mu_{S'}, D_{S'}) \otimes \delta_{S'^c}}{\hat{w}_{S'}d\Pi_{S'}(\cdot|Y) \otimes \delta_{S'^c}}(\theta)$$
$$= \log(1/\hat{w}_{S'}) + \mathrm{KL}(N_{S'}(\mu_{S'}, D_{S'})||\Pi_{S'}(\cdot|Y)),$$

where the last KL divergence is over distributions in $\mathbb{R}^{|S'|}$. On $\mathcal{A}_n$, $\log(1/\hat{w}_{S'}) \leq \log(2ep^{Ls_0}) = \log(2e) + Ls_0 \log p$ by (24). Writing $E_{\mu_{S'}, D_{S'}}$ for the expectation under the law $\theta_{S'} \sim N_{S'}(\mu_{S'}, D_{S'})$,

$$\mathrm{KL}(N_{S'}(\mu_{S'}, D_{S'})||\Pi_{S'}(\cdot|Y)) = E_{\mu_{S'}, D_{S'}} \left[ \log \frac{dN_{S'}(\mu_{S'}, D_{S'})}{dN_{S'}(\mu_{S'}, \Sigma_{S'})}(\theta_S) + \log \frac{dN_{S'}(\mu_{S'}, \Sigma_{S'})}{d\Pi_{S'}(\cdot|Y)}(\theta_S) \right], \tag{25}$$

where again $\Sigma_{S'} = (X_{S'}^T X_{S'})^{-1}$. Using the formula for the KL divergence between two multivariate Gaussian distributions, the first term in the last display equals

$$\mathrm{KL}(N_{S'}(\mu_{S'}, D_{S'})||N_{S'}(\mu_{S'}, \Sigma_{S'})) = \frac{1}{2}\left(\log \frac{\det \Sigma_{S'}}{\det D_{S'}} - |S'| + \mathrm{Tr}(\Sigma_{S'}^{-1} D_{S'})\right).$$

Using the definitions (22)-(23) gives $\mathrm{Tr}(\Sigma_{S'}^{-1} D_{S'}) = |S'|$. Turning to the determinants,

$$\det D_{S'}^{-1} = \prod_{j=1}^{|S'|}(X_{S'}^T X_{S'})_{jj} = \prod_{j\in S'} \|X_{\cdot j}\|_2^2 \leq \|X\|^{2|S'|}.$$

Let $\Lambda_{\max}(A)$ and $\Lambda_{\min}(A)$ denote the largest and smallest eigenvalues, respectively, of a square matrix $A$. Recall the diagonal matrix $W$ from (7), whose entries satisfy $W_{ii} \in (0, 1/4]$. Using the variational characterization of the minimal eigenvalue of a symmetric matrix ([9], p234),

$$\Lambda_{\min}(X_{S'}^T X_{S'}) = \min_{u\in\mathbb{R}^{|S'|}:u\neq 0} \frac{u^T X_{S'}^T X_{S'} u}{\|u\|_2^2} \geq 4 \min_{v\in\mathbb{R}^p:v\neq 0, v_{S'^c}=0} \frac{v^T X^T WXv}{\|v\|_2^2} \geq 4\underline{\kappa}(|S'|)\|X\|^2. \tag{26}$$

Since $\Sigma_{S'} = (X_{S'}^T X_{S'})^{-1}$ is positive definite, the last display implies

$$\det \Sigma_{S'} \leq \Lambda_{\max}((X_{S'}^T X_{S'})^{-1})^{|S'|} = (1/\Lambda_{\min}(X_{S'}^T X_{S'}))^{|S'|} \leq \frac{1}{(4\underline{\kappa}(|S'|)\|X\|^2)^{|S'|}}.$$

Combining these bounds,

$$\mathrm{KL}(N_{S'}(\mu_{S'}, D_{S'}) || N_{S'}(\mu_{S'}, \Sigma_{S'})) = \tfrac{1}{2}\log(\det \Sigma_{S'} \det D_{S'}^{-1})$$
$$\leq |S'|\log(1/(4\underline{\kappa}(|S'|)))$$
$$\leq L s_0 \log(1/(4\underline{\kappa}(Ls_0))).$$

It thus remains to bound the second term in (25), which has non-diagonal covariance matrix $\Sigma_{S'}$.

**Bounding the non-diagonal covariance case.** One can check that $-\tfrac{1}{2}(\theta_{S'} - \mu_{S'})^T \Sigma_{S'}^{-1}(\theta_{S'} - \mu_{S'})$ equals

$$-\tfrac{1}{2}(\theta_{S'} - \theta_{0,S'})^T X_{S'}^T X_{S'}(\theta_{S'} - \theta_{0,S'}) + (Y - E_{\theta_0}Y)^T X_{S'}(\theta_{S'} - \theta_{0,S'}) + C_{S'}(X, Y),$$

where $C_{S'}(X, Y)$ does not depend on $\theta$. Let $\bar\theta_{S'}$ denote the extension of a vector $\theta_{S'} \in \mathbb{R}^{|S'|}$ to $\mathbb{R}^p$ with $\bar\theta_{S',j} = \theta_{S,j}$ for $j \in S'$ and $\bar\theta_{S',j} = 0$ for $j \notin S'$. Since $(Y - E_{\theta_0}Y)^T X_{S'}(\theta_{S'} - \theta_{0,S'}) = \nabla_\theta \ell_{n,\theta_0}(Y)^T(\bar\theta_{S'} - \bar\theta_{0,S'})$, the density function of the $N_{S'}(\mu_{S'}, \Sigma_{S'})$ distribution is thus proportional to $e^{-\frac{1}{2}\|X_{S'}(\theta_{S'} - \theta_{0,S'})\|_2^2 + \nabla_\theta \ell_{n,\theta_0}(Y)^T(\bar\theta_{S'} - \bar\theta_{0,S'})}$, $\theta_{S'} \in \mathbb{R}^{|S'|}$. Using Bayes formula and the Taylor expansion (20), $\Pi_{S'}(\cdot | Y)$ has density proportional to

$$\exp\left(\ell_{n,\bar\theta_{S'}}(Y) - \ell_{n,\theta_0}(Y) - \lambda\|\theta_{S'}\|_1\right)$$
$$\propto \exp\left(\nabla_\theta \ell_{n,\theta_0}(Y)^T(\bar\theta_{S'} - \bar\theta_{0,S'}) + \mathcal{L}_{n,\bar\theta_{S'}}(Y) - \lambda\|\theta_{S'}\|_1\right).$$

Using these representations of the two densities, the second term in (25) can be rewritten as

$$E_{\mu_{S'}, D_{S'}}\left[\log \frac{D_\Pi e^{-\frac{1}{2}\|X_{S'}(\theta_{S'} - \theta_{0,S'})\|_2^2 + \nabla_\theta \ell_{n,\theta_0}(Y)^T(\bar\theta_{S'} - \bar\theta_{0,S'}) - \lambda\|\theta_{0,S'}\|_1}}{D_N e^{\nabla_\theta \ell_{n,\theta_0}(Y)^T(\bar\theta_{S'} - \bar\theta_{0,S'}) + \mathcal{L}_{n,\bar\theta_{S'}}(Y) - \lambda\|\theta_{S'}\|_1}}\right]$$
$$= E_{\mu_{S'}, D_{S'}}\left[-\tfrac{1}{2}\|X_{S'}(\theta_{S'} - \theta_{0,S'})\|_2^2 - \mathcal{L}_{n,\bar\theta_S}(Y)\right]$$
$$+ \lambda E_{\mu_{S'}, D_{S'}}(\|\theta_{S'}\|_1 - \|\theta_{0,S'}\|_1) + \log(D_\Pi / D_N)$$
$$=: (I) + (II) + (III),$$

where the normalizing constants are $D_\Pi = \int_{\mathbb{R}^{|S'|}} e^{\nabla_\theta \ell_{n,\theta_0}(Y)^T(\bar\theta_{S'} - \bar\theta_{0,S'}) + \mathcal{L}_{n,\bar\theta_{S'}}(Y) - \lambda\|\theta_{S'}\|_1} d\theta_{S'}$ and $D_N = \int_{\mathbb{R}^{|S'|}} e^{-\frac{1}{2}\|X_{S'}(\theta_{S'} - \theta_{0,S'})\|_2^2 + \nabla_\theta \ell_{n,\theta_0}(Y)^T(\bar\theta_{S'} - \bar\theta_{0,S'}) - \lambda\|\theta_{0,S'}\|_1} d\theta_{S'}$. We now bound $(I) - (III)$ in turn.

$(I)$: Using the likelihood (19) and the mean-value form of the remainder in the Taylor expansion (20), for $\xi_i$ between $x_i^T \bar\theta_{S'}$ and $x_i^T \theta_0$,

$$\mathcal{L}_{n,\bar\theta_{S'}} = -\frac{1}{2}\sum_{i=1}^n g''(\xi_i)|x_i^T(\bar\theta_{S'} - \theta_0)|^2$$
$$\geq -\frac{1}{8}\sum_{i=1}^n 2|x_i^T(\bar\theta_{S'} - \bar\theta_{0,S'})|^2 + 2|x_i^T(\bar\theta_{0,S'} - \theta_0)|^2 \qquad (27)$$
$$= -\frac{1}{4}\|X_{S'}(\theta_{S'} - \theta_{0,S'})\|_2^2 - \frac{1}{4}\|X_{S'^c}\theta_{0,S'^c}\|_2^2,$$

so that $(I)$ is bounded by $\|X_{S'^c}\theta_{0,S'^c}\|_2^2/4$. On $\mathcal{A}_n$, using (24),

$$\|X_{S'^c}\theta_{0,S'^c}\|_2^2 = \|X\bar\theta_{0,S'^c}\|_2^2 \leq \overline{\kappa}(|S_0 \cap S'^c|)\|X\|^2\|\theta_{0,S'^c}\|_2^2 \leq \overline{\kappa}(s_0)M_1^2 s_0 \log p,$$

so that $(I) \leq \overline{\kappa}(s_0)M_1^2 s_0 \log p/4$.

$(II)$: Under the expectation $E_{\mu_{S'}, D_{S'}}$, we have the equality in distribution $\theta_{S'} - \theta_{0,S'} =^d (X_{S'}^T X_{S'})^{-1} X_{S'}^T(Y - E_{\theta_0}Y) + Z$, where $Z \sim N_{S'}(0, D_{S'})$. Applying the triangle inequality and Cauchy-Schwarz,

$$(II) \leq \lambda\|(X_{S'}^T X_{S'})^{-1} X_{S'}^T(Y - E_{\theta_0}Y)\|_1 + \lambda E_{\mu_{S'}, D_{S'}}\|Z\|_1$$
$$\leq \lambda|S'|^{1/2}(\|(X_{S'}^T X_{S'})^{-1} X_{S'}^T(Y - E_{\theta_0}Y)\|_2 + \mathrm{Tr}(D_{S'})^{1/2}),$$

since by Jensen's inequality $E\|Z\|_2 \le (E\|Z\|_2^2)^{1/2} = \mathrm{Tr}(D_{S'})^{1/2}$. Using the definition (23), for $e_j$ the $j^{th}$ unit vector in $\mathbb{R}^p$,

$$\mathrm{Tr}(D_{S'}) = \sum_{j=1}^{|S'|} \frac{1}{(X_{S'}^T X_{S'})_{jj}} = \sum_{j \in S'} \frac{1}{\|Xe_j\|_2^2} \le \sum_{j \in S'} \frac{1}{\underline{\kappa}(1)\|X\|^2} = \frac{|S'|}{\underline{\kappa}(1)\|X\|^2}.$$

The matrix operator norm (from $\mathbb{R}^{|S'|}$ to $\mathbb{R}^{|S'|}$) of $(X_{S'}^T X_{S'})^{-1}$ equals its largest eigenvalue, which is bounded by $1/(4\underline{\kappa}(|S'|)\|X\|^2)$ using (26). Recalling that $\nabla_\theta \ell_{n,\theta_0}(Y) = X^T(Y - E_{\theta_0}Y)$, on the event $\mathcal{A}_n$ the first term in the second to last display is therefore bounded by

$$\frac{\lambda|S'|^{1/2}}{4\underline{\kappa}(|S'|)\|X\|^2}\|X_{S'}^T(Y - E_{\theta_0}Y)\|_2 \le \frac{\lambda|S'|}{4\underline{\kappa}(|S'|)\|X\|^2}\|X^T(Y - E_{\theta_0}Y)\|_\infty \le \frac{\lambda L s_0 t}{4\underline{\kappa}(Ls_0)\|X\|^2}.$$

We have thus shown that

$$(II) \le \frac{\lambda L s_0}{\|X\|}\left(\frac{t}{4\underline{\kappa}(Ls_0)\|X\|} + \frac{1}{\underline{\kappa}(1)^{1/2}}\right).$$

$(III)$: It remains to control the ratio of normalizing constants $\log(D_\Pi/D_N)$. Define

$$B_{S'} = \{\theta_{S'} \in \mathbb{R}^{|S'|} : \|\theta_{S'} - \theta_{0,S'}\|_2 \le 2M_1\sqrt{s_0 \log p}/\|X\|\}.$$

On $\mathcal{A}_n$, using (21) and (24),

$$\Pi_{S'}(B_{S'}^c|Y) \le \frac{\hat{w}_{S'}}{\hat{w}_{S'}}\Pi_{S'}(\theta_{S'} \in \mathbb{R}^{|S'|} : \|\bar{\theta}_{S'} - \theta_0\|_2 > 2M_1\sqrt{s_0 \log p}/\|X\| - \|\theta_{0,S'^c}\|_2|Y)$$

$$\le \hat{w}_{S'}^{-1}\Pi(\theta \in \mathbb{R}^p : \|\theta - \theta_0\|_2 > M_1\sqrt{s_0 \log p}/\|X\|\,|Y)$$

$$\le 2ep^{Ls_0}e^{-M_2 s_0 \log p} = 2e^{1-(M_2-L)s_0 \log p} \le 1/2,$$

where the last inequality follows from rearranging the assumption $(4e)^{1/(M_2-L)} \le p^{s_0}$. Using Bayes formula, this gives

$$\Pi_{S'}(B_{S'}|Y)1_{\mathcal{A}_n} = \frac{\int_{B_{S'}} e^{\ell_{n,\bar{\theta}_{S'}}(Y) - \ell_{n,\theta_0}(Y) - \lambda\|\theta_{S'}\|_1}d\theta_{S'}}{\int_{\mathbb{R}^{|S'|}} e^{\ell_{n,\bar{\theta}_{S'}}(Y) - \ell_{n,\theta_0}(Y) - \lambda\|\theta_{S'}\|_1}d\theta_{S'}}1_{\mathcal{A}_n} \ge \frac{1}{2}1_{\mathcal{A}_n}.$$

By (20) the denominator in the last display equals $e^{\nabla_\theta \ell_{n,\theta_0}(Y)^T(\bar{\theta}_{0,S'}-\theta_0)}D_\Pi$, which implies that on $\mathcal{A}_n$, $D_\Pi \le 2\int_{B_{S'}} e^{\nabla_\theta \ell_{n,\theta_0}(Y)^T(\bar{\theta}_{S'}-\bar{\theta}_{0,S'})+\mathcal{L}_{n,\bar{\theta}_{S'}}(Y)-\lambda\|\theta_{S'}\|_1}d\theta_{S'}$. Thus on $\mathcal{A}_n$,

$$\log\frac{D_\Pi}{D_N} \le \log\frac{2\int_{B_{S'}} e^{\nabla_\theta \ell_{n,\theta_0}(Y)^T(\bar{\theta}_{S'}-\bar{\theta}_{0,S'})+\mathcal{L}_{n,\bar{\theta}_{S'}}(Y)-\lambda\|\theta_{S'}\|_1}d\theta_{S'}}{\int_{B_{S'}} e^{-\frac{1}{2}\|X_{S'}(\theta_{S'}-\theta_{0,S'})\|_2^2+\nabla_\theta \ell_{n,\theta_0}(\bar{\theta}_{S'}-\bar{\theta}_{0,S'})-\lambda\|\theta_{0,S'}\|_1}d\theta_{S'}}$$

$$\le \log\left(\sup_{\theta_{S'}\in B_{S'}} e^{\mathcal{L}_{n,\bar{\theta}_{S'}}(Y)+\frac{1}{2}\|X_{S'}(\theta_{S'}-\theta_{0,S'})\|_2^2+\lambda\|\theta_{0,S'}\|_1-\lambda\|\theta_{S'}\|_1}\right) + \log 2$$

$$\le \sup_{\theta_{S'}\in B_{S'}} \mathcal{L}_{n,\bar{\theta}_{S'}}(Y) + \frac{1}{2}\|X_{S'}(\theta_{S'}-\theta_{0,S'})\|_2^2 + \lambda\|\theta_{S'}-\theta_{0,S'}\|_1 + \log 2.$$

Now $\mathcal{L}_{n,\bar{\theta}_{S'}}(Y) < 0$ by (27) since $g'' > 0$. Using Cauchy-Schwarz and the definition of $B_{S'}$, on $\mathcal{A}_n$,

$$(III) = \log\frac{D_\Pi}{D_N} \le \sup_{\theta_{S'}\in B_{S'}} \frac{1}{2}\overline{\kappa}(|S'|)\|X\|^2\|\theta_{S'}-\theta_{0,S'}\|_2^2 + \lambda|S'|^{1/2}\|\theta_{S'}-\theta_{0,S'}\|_2 + \log 2$$

$$\le 2\overline{\kappa}(Ls_0)M_1^2 s_0 \log p + \frac{2M_1\lambda L^{1/2}s_0\sqrt{\log p}}{\|X\|} + \log 2.$$

Combining all of the above bounds gives the result. $\qquad\square$

## 10.2 Contraction results

The second part to applying Lemma 1 is showing that on an event, the desired sets have all but exponentially small posterior probability. This involves using results on dimension selection and posterior contraction from high-dimensional Bayesian statistics, especially Atchadé [1] and Castillo et al. [7]. The following results follow closely the proofs in [1], but we reproduce them here for convenience, since in that paper they are not stated or proved in the exponential form needed to apply Lemma 1. We are also able to simplify certain technical conditions and streamline some proofs. Note that his results, including the definitions of the compatibility constants, match when $\|X\| \sim \sqrt{n}$.

We next introduce some notation from [1], used throughout this section. A continuous function $r : [0, \infty) \to [0, \infty)$ is called a *rate function* if it is strictly increasing, $r(0) = 0$ and $\lim_{x \downarrow 0} r(x)/x = 0$. For a rate function $r$ and $a \geq 0$, define

$$\phi_r(a) = \inf\{x > 0 : r(z) \geq az, \text{ for all } z \geq x\}, \tag{28}$$

with the convention $\inf \emptyset = \infty$. Let $B(\Theta, M) = \{\theta \in \theta_0 + \Theta : \|\theta - \theta_0\|_2 \leq M\}$ denote the $\ell_2$-ball of radius $M > 0$ centered at $\theta_0$ with elements in $\theta_0 + \Theta$. For $\varepsilon > 0$, we denote by $D(\varepsilon, B(\Theta, M))$ the $\varepsilon$-packing number of $B(\Theta, M)$, namely the maximal number of points in $B(\Theta, M)$ such that the $\ell_2$ distance between any two points is at least $\varepsilon$.

The following result bounds the posterior probability of selecting a model of size larger than a multiple of the true model size.

**Lemma 8** (Theorem 4(1) of [1]). *Suppose the prior satisfies* (3) *and* (4), $p^{A_4} \geq 8A_2$, *and that* $\|X\| \geq (64/3)\alpha s_0 \sqrt{\log p}/\underline{\kappa}$. *Then for any $L > 1$,*

$$E_{\theta_0}[\Pi\left(\theta \in \mathbb{R}^p : |S_\theta| \geq Ls_0 \mid Y\right) 1_{\mathcal{A}_{n,1}(\lambda/2)}] \leq 2 \exp\left(-s_0 \log p\left[L\left(A_4 - \frac{\log(4A_2)}{\log p}\right) - C\right]\right),$$

*where $C = 1 + \frac{4\alpha^2}{\underline{\kappa}} + \log(4 + \overline{\kappa}(s_0)/\log p) + (1 + \frac{1}{s_0})(A_4 - \frac{\log(4A_2)}{\log p})$.*

*Proof.* By Lemma 12(1), for any $k \geq 0$,

$$E_{\theta_0}[\Pi(\theta : |S_\theta| \geq s_0 + k|Y)1_{\mathcal{A}_{n,1}(\lambda/2)}] \leq 2e^a \left(4 + \frac{\overline{\kappa}(s_0)\|X\|^2}{\lambda^2}\right)^{s_0} \binom{p}{s_0}\left(\frac{4A_2}{p^{A_4}}\right)^k,$$

where $a = -\frac{1}{2}\inf_{x>0}[\frac{\underline{\kappa}\|X\|^2 x^2}{1+4s_0^{1/2}\|X\|_\infty x} - 4\lambda s_0^{1/2}x]$. It remains to simplify the right-hand side.

One can check that for $\tau, b, c > 0$, $\inf_{x>0}[\frac{\tau x^2}{1+bx} - cx] \geq -\frac{c^2}{4\tau^{1/2}(\tau - cb)^{1/2}} \geq -\frac{c^2}{2\tau}$ if $\tau \geq 4bc/3$. In our setting, this condition equals $\|X\|^2 \underline{\kappa} \geq (64/3)\lambda s_0 \|X\|_\infty$, which holds by assumption. This yields $a \leq \frac{4\lambda^2 s_0}{\|X\|^2 \underline{\kappa}}$. Using the upper bound $\binom{p}{s_0} \leq p^{s_0}$, setting $k = \lfloor(L-1)s_0\rfloor$ and using that $(L-1)s_0 - 1 \leq \lfloor(L-1)s_0\rfloor \leq (L-1)s_0$ gives the result. $\square$

The next result is the analogous version of Theorem 4(2) in [1] with the exponential bounds we require here. It provides a contraction rate for posterior models of a given size.

**Lemma 9.** *Suppose the prior satisfies* (3) *and* (4), $p^{A_4} \geq 8A_2$, *and that* $\|X\| \geq 50\alpha(L+2)s_0\sqrt{\log p}\|X\|_\infty/\underline{\kappa}((L+1)s_0)$ *for some $L > 0$. Then for any $\theta_0 \in \mathbb{R}^p$, and $M \geq \max(25\alpha, (1+A_3)/16)$,*

$$E_{\theta_0}\left[\Pi(\theta \in \mathbb{R}^p : |S_\theta| \leq Ls_0, \|\theta - \theta_0\|_2 \geq \frac{8M\sqrt{(L+2)s_0 \log p}}{\underline{\kappa}((L+1)s_0)\|X\|}|Y)1_{\mathcal{A}_{n,1}(\|X\|\sqrt{\log p})}\right]$$
$$\leq 6e^{-s_0 \log p(8LM - C_L)},$$

*where $C_L = \max(L(1 + \frac{\log(24)}{\log p}), \tilde{C}_p)$ and $\tilde{C}_p = \frac{\log A_1 + \log(1 + 4\alpha^2 \log p)}{\log p}$.*

While $\tilde{C}_p$ is not a true constant since it depends on $p$, we write it as such since it is asymptotically negligible. As $p \to \infty$, we have $\tilde{C}_p \to 0$ and $C_L \to L$.

*Proof.* We write $\underline{\kappa}_L = \underline{\kappa}((L+1)s_0)$ during this proof to ease notation. By Lemma 12(2), for any $M > 2$,

$$E_{\theta_0}\Pi(\theta \in \theta_0 + \bar{\Theta}_L : \|\theta - \theta_0\|_2 > M\varepsilon|Y)1_{\mathcal{A}_{n,1}(\|X\|\sqrt{\log p})}$$

$$\leq \sum_{j\geq 1} D_j e^{-r(\frac{jM\varepsilon}{2})/8} + 2\binom{p}{s_0}\left(\frac{p^{A_3}}{A_1}\right)^{s_0}\left(1 + \frac{4\lambda^2}{\bar{\kappa}(s_0)\|X\|^2}\right)^{s_0}\sum_{j\geq 1} e^{-r(\frac{jM\varepsilon}{2})/8}e^{3\lambda c_0 jM\varepsilon}, \quad (29)$$

where the quantities in (29) are defined in that lemma. Note that for $\theta$ satisfying $|S_\theta| \leq Ls_0$, then $|S_{\theta-2\theta_0}| \leq (L+1)s_0$, so that $\theta - \theta_0 \in \bar{\Theta}_L$. We may thus further restrict the set in the last display to $\{\theta : |S_\theta| \leq Ls_0\}$, as in the posterior probability in the lemma. We first compute $\varepsilon$ and then simplify the right-hand side of (29).

Recall that we may take as rate any $\varepsilon \geq \phi_r(2\eta_L)$ for $r$ the rate function in Lemma 12(2). For a rate function $r(x) = \frac{\tau x^2}{1+bx}, \tau, b > 0$, the inequality $r(x) \geq ax$ is equivalent to $x((\tau-ab)x-a) \geq 0$. Using the definition (28) thus gives $\phi_r(a) = \frac{a}{\tau-ab}$. Setting $\tau = \underline{\kappa}_L\|X\|^2$ and $b = \|X\|_\infty\sqrt{(L+1)s_0}/2$ as in Lemma 12(2), and using our assumption $\frac{1}{2}\|X\|^2\underline{\kappa}_L \geq 25\alpha(L+2)s_0\|X\|\sqrt{\log p}\|X\|_\infty \geq \eta_L\sqrt{(L+1)s_0}\|X\|_\infty$, we get

$$\phi_r(2\eta_L) = \frac{2\eta_L}{\underline{\kappa}_L\|X\|^2 - \eta_L\|X\|_\infty\sqrt{(L+1)s_0}} \leq \frac{4\eta_L}{\underline{\kappa}_L\|X\|^2} = \frac{8\sqrt{(L+2)s_0\log p}}{\underline{\kappa}_L\|X\|} =: \varepsilon$$

Turning to the right-hand side of (29), note that $c_0 = \sup_{v\in\bar{\Theta}_L}\|v\|_1/\|v\|_2 \leq \sqrt{(L+2)s_0}$. Arguing as on p. 29-30 of [1] gives $\sum_{j\geq 1} D_j e^{-r(jM\varepsilon/2)/8} \leq 2\exp\left((L+2)s_0\log p[1 + \frac{\log(24)}{\log p} - 8M]\right)$. Similarly, setting $x = jM\varepsilon/2$,

$$3\lambda\sqrt{(L+2)s_0}jM\varepsilon - \frac{1}{8}r(jM\varepsilon/2) = -\frac{x}{8}\left(\frac{\|X\|^2\underline{\kappa}_L x}{1 + \frac{1}{2}\sqrt{(L+1)s_0}\|X\|_\infty x} - 48\lambda\sqrt{(L+2)s_0}\right)$$

$$\leq -\frac{x}{8}\left(\frac{\|X\|^2\underline{\kappa}_L\frac{M\varepsilon}{2}}{1 + \frac{1}{2}\sqrt{(L+1)s_0}\|X\|_\infty\frac{M\varepsilon}{2}} - 48\lambda\sqrt{(L+2)s_0}\right)$$

$$\leq -\frac{\lambda\sqrt{(L+2)s_0}x}{4}$$

as long as

$$\frac{\|X\|^2\underline{\kappa}_L\frac{M\varepsilon}{2}}{1 + \frac{1}{2}\sqrt{(L+1)s_0}\|X\|_\infty\frac{M\varepsilon}{2}} \geq 50\lambda\sqrt{(L+2)s_0}.$$

We show the last display holds under the present assumptions. Since $\sqrt{(L+1)s_0}\|X\|_\infty\varepsilon \leq \frac{8(L+2)s_0\sqrt{\log p}\|X\|_\infty}{\|X\|\underline{\kappa}_L} \leq 8/(50\alpha)$ by assumption, the left-hand side is lower bounded by $\frac{\|X\|^2\underline{\kappa}_L M\varepsilon}{2+4M/(50\alpha)} \geq (50\alpha/8)\|X\|^2\underline{\kappa}_L\varepsilon$ for $M \geq 25\alpha$. Since $\lambda \leq \alpha\|X\|\sqrt{\log p}$ by assumption, the last display holds following from

$$\frac{(50\alpha/8)\|X\|^2\underline{\kappa}_L\varepsilon}{50\sqrt{(L+2)s_0}} = \alpha\|X\|\sqrt{\log p}.$$

This implies

$$\sum_{j\geq 1} e^{-\frac{1}{8}r(jM\varepsilon/2)}e^{3\lambda c_0 jM\varepsilon} \leq \sum_{j\geq 1} e^{-\frac{jM\lambda\sqrt{(L+2)s_0}\varepsilon}{8}} \leq 2e^{-8M(L+2)s_0\log p},$$

where the last inequality again follows by the same argument on p. 29-30 of [1].

Summing up these bounds and using $\binom{p}{s_0} \leq p^{s_0}$ and $\bar{\kappa}(s_0) \geq \bar{\kappa}(1) = 1$, the right-hand side of (29) is bounded by

$$2\exp\left((L+2)s_0\log p\left[1 + \frac{\log(24)}{\log p} - 8M\right]\right)$$

$$+ 4\exp\left((1+A_3)s_0\log p + s_0\log\left(1 + 4\alpha^2\log p\right) - s_0\log A_1 - 8M(L+2)s_0\log p\right)$$

$$\leq 6\exp\left(-s_0\log p\left[8LM - \max(L(1 + \frac{\log(24)}{\log p}), \tilde{C}_p))\right]\right).$$

$\square$

Combining the last two lemmas yields the contraction rate result with exponential bounds.

**Lemma 10.** *Suppose the prior satisfies (3) and (4), and $p^{A_4} \geq 8A_2$. If for $K > 0$, the design matrix satisfies condition (8) with $L = L_K = \frac{K+C}{A_4 - \log(4A_2)/\log p}$, with $C$ the constant in Lemma 8, then for any $\theta_0 \in \mathbb{R}^p$,*

$$E_{\theta_0}\left[\Pi\left(\theta \in \mathbb{R}^p : \|\theta - \theta_0\|_2 \geq C_K \frac{\sqrt{s_0 \log p}}{\|X\|}\Big| Y\right) 1_{\mathcal{A}_{n,1}(\|X\|\sqrt{\log p})}\right] \leq 8e^{-Ks_0 \log p},$$

*where $C_K = \frac{8M_K\sqrt{L_K+2}}{\underline{\kappa}((L_K+1)s_0)}$, $M_K = \max(25\alpha, \frac{1+A_3}{16}, \frac{K+\tilde{C}_p}{8L_K} + \frac{1}{8} + \frac{\log(24)}{8\log p})$ and $\tilde{C}_p$ is given in Lemma 9.*

If all the compatibility constants are bounded away from zero and infinity, the constants in Lemma 10 scale like $L_K \sim K$, $M_K \sim \sqrt{K}$ and $C_K \sim K$ as $K \to \infty$. We now have the required event to apply Lemma 1.

**Lemma 11.** *Suppose the prior satisfies (3) and (4), and $p^{A_4} \geq 8A_2$. Set $L = \frac{1+C}{A_4 - \log(4A_2)/\log p}$, where $C$ is the constant in Lemma 8, and assume the design matrix satisfies (8) for this $L$. Then for $t = \|X\|\sqrt{\log p}$, $M_2 = 2L$ and $M_1 = C_{3L}$, where $C_{3L}$ is the constant in Lemma 10 with $K = 3L$, and any $\theta_0 \in \mathbb{R}^p$,*

$$P_{\theta_0}\left(\mathcal{A}_n(t, L, M_1, M_2)^c\right) \leq 2/p + (8/3)p^{-s_0} + 8p^{-Ls_0},$$

*where $\mathcal{A}_n(t, L, M_1, M_2)$ is defined in (15).*

*Proof.* Using a union bound and the definition (15),

$$P_{\theta_0}\left(\mathcal{A}_n(t, L, M_1, M_2)^c\right) \leq P_{\theta_0}(\mathcal{A}_{n,1}(\|X\|\sqrt{\log p})^c) + P_{\theta_0}(\mathcal{A}_{n,2}(L)^c \cap \mathcal{A}_{n,1}(\|X\|\sqrt{\log p}))$$
$$+ P_{\theta_0}(\mathcal{A}_{n,3}(M_1, M_2)^c \cap \mathcal{A}_{n,1}(\|X\|\sqrt{\log p})).$$

Since $\frac{\partial}{\partial \theta_j}\ell_{n,\theta_0}(Y) = \sum_{i=1}^n (Y_i - g'(x_i^T\theta_0))X_{ij}$, by Hoeffding's inequality,

$$P_{\theta_0}(\mathcal{A}_{n,1}(t)^c) = P_{\theta_0}\left(\max_{1\leq j\leq p}\left|\sum_{i=1}^n (Y_i - g'(x_i^T\theta_0))X_{ij}\right| > t\right)$$

$$\leq 2\sum_{j=1}^p e^{-\frac{2t^2}{\|X_{\cdot j}\|_2^2}} \leq 2pe^{-\frac{2t^2}{\|X\|^2}} = \frac{2}{p}.$$

Applying Markov's inequality and Lemma 8 with the present choice of $L$, the second term is bounded by

$$(4/3)E_{\theta_0}\left[\Pi(\theta \in \mathbb{R}^p : |S_\theta| > Ls_0|Y)1_{\mathcal{A}_{n,1}(\|X\|\sqrt{\log p})}\right] \leq (8/3)e^{-s_0 \log p}.$$

Similarly, using Markov's inequality and Lemma 10 with $K = 3L$, the third term is bounded by

$$e^{M_2 s_0 \log p}E_{\theta_0}\left[\Pi(\theta \in \mathbb{R}^p : \|\theta - \theta_0\|_2 \geq M_1\sqrt{s_0 \log p}/\|X\||Y)1_{\mathcal{A}_{n,1}(\|X\|\sqrt{\log p})}\right] \leq 8e^{-Ls_0 \log p}.$$

$\square$

The following is a simplified version of Theorem 3 of Atchadé [1], which applies to general settings, tailored to the sparse high-dimensional logistic regression model. It gives high level technical conditions under which one can control (1) the posterior model dimension and (2) the posterior $\ell_2$ norm for models of restricted dimension.

**Lemma 12.** *Suppose the prior satisfies (3) and $p^{A_4} \geq 8A_2$.*

*(1) For any integer $k \geq 0$,*

$$E_{\theta_0}\Pi\left(\theta \in \mathbb{R}^p : |S_\theta| \geq s_0 + k|Y\right)1_{\mathcal{A}_{n,1}(\lambda/2)} \leq 2e^a\left(4 + \frac{\overline{\kappa}(s_0)\|X\|^2}{\lambda^2}\right)^{s_0}\binom{p}{s_0}\left(\frac{4A_2}{p^{A_4}}\right)^k,$$

*where*

$$a = -\frac{1}{2}\inf_{x>0}\left[\frac{\underline{\kappa}\|X\|^2 x^2}{1 + 4s_0^{1/2}\|X\|_\infty x} - 4\lambda\sqrt{s_0}x\right].$$

(2) For $L > 0$, set $\bar{\Theta}_L = \{\theta \in \mathbb{R}^p : |S_{\theta-\theta_0}| \leq (L+1)s_0\}$ *and define the rate function* $r(x) = \frac{\kappa((L+1)s_0)\|X\|^2 x^2}{1+\|X\|_\infty \sqrt{(L+1)s_0 x/2}}$. *Further set* $\eta_L = 2\sqrt{(L+2)s_0}\|X\|\sqrt{\log p}$ *and* $\varepsilon = \phi_r(2\eta_L)$, *where* $\phi_r$ *uses the same rate function* $r$ *and is defined in* (28). *Then for any* $M_0 > 2$,

$$E_{\theta_0}\Pi(\theta \in \theta_0 + \bar{\Theta}_L : \|\theta - \theta_0\|_2 > M_0\varepsilon|Y)1_{\mathcal{A}_{n,1}(\|X\|\sqrt{\log p})}$$

$$\leq \sum_{j\geq 1} D_j e^{-r(\frac{jM_0\varepsilon}{2})/8} + 2\binom{p}{s_0}\Big(\frac{p^{A_3}}{A_1}\Big)^{s_0}\Big(1 + \frac{4\lambda^2}{\bar{\kappa}(s_0)\|X\|^2}\Big)^{s_0} \sum_{j\geq 1} e^{-r(\frac{jM_0\varepsilon}{2})/8}e^{3\lambda c_0 j M_0\varepsilon},$$

*where* $c_0 = \sup_{u\in\bar{\Theta}_L}\sup_{v\in\bar{\Theta}_L,\|v\|_2=1}|\langle sign(u),v\rangle|$ *and* $D_j = D\big(\frac{jM_0\varepsilon}{2}, B(\bar{\Theta}_L,(j+1)M_0\varepsilon)\big)$.

*Proof.* This is a combination of Theorems 3 and 4 in [1]. In particular, we verify that certain technical assumptions of that result hold automatically in the logistic regression model, giving the simpler result above. Firstly note that Assumptions H1-H3 of [1] are satisfied (H1) by definition, (H2) since $\theta \mapsto \ell_{n,\theta}$ in (19) is concave and differentiable and (H3) by (3).

For $y \in \{0,1\}^n$ data in model (1), $\mathcal{L}_{n,\theta}$ defined in (20), $\Theta_0 = \{\theta : S_\theta \subseteq S_{\theta_0}\}$ and $r$ some rate function, define

$$\mathcal{N} = \left\{\theta \in \mathbb{R}^p : \theta \neq 0, \text{ and } \sum_{i\in S_0^c}|\theta_i| \leq 7\|\theta_{S_0}\|_1\right\},$$

$$\check{\mathcal{E}}_{n,1}(\mathcal{N},r) = \left\{y \in \{0,1\}^n : \forall\theta \in \theta_0 + \mathcal{N} : \mathcal{L}_{n,\theta}(y) \leq -\tfrac{1}{2}r(\|\theta-\theta_0\|_2)\right\},$$

$$\hat{\mathcal{E}}_{n,1}(\Theta_0,\bar{L}) = \left\{y \in \{0,1\}^n : \forall\theta \in \theta_0 + \Theta_0, \mathcal{L}_{n,\theta}(y) \geq -\tfrac{\bar{L}}{2}\|\theta-\theta_0\|_2^2\right\},$$

$$\mathcal{E}_{n,0}(\Theta,\lambda) = \left\{y \in \{0,1\}^n : \sup_{u\in\Theta,\|u\|_2=1}|\langle\nabla_\theta\log\ell_{n,\theta_0}(y),u\rangle| \leq \tfrac{\lambda}{2}\right\},$$

where $\bar{L} > 0$ and $\lambda > 0$ is the regularization parameter in the prior (2). This matches the notation in [1] (except note his $\rho$ is our $\lambda$), where it is shown the theorem's two conclusions hold under various choices of parameters in the last displays.

Part (1): [1] considers the event $\mathcal{E}_{n,0}(\mathbb{R}^p,\lambda) \cap \hat{\mathcal{E}}_{n,1}(\Theta_0,\bar{L}) \cap \check{\mathcal{E}}_n(\mathcal{N},r)$, which we now simplify. Arguing as on p27 of [1] yields

$$\mathcal{L}_{n,\theta}(y) \leq -\sum_{i=1}^n g''(x_i^T\theta_0)\frac{|x_i^T(\theta-\theta_0)|^2}{2+|x_i^T(\theta-\theta_0)|},$$

For $\theta - \theta_0 \in \mathcal{N}$,

$$|x_i^T(\theta-\theta_0)| \leq \|X\|_\infty\|\theta-\theta_0\|_1 \leq 8\|X\|_\infty s_0^{1/2}\|\theta-\theta_0\|_2,$$

which gives

$$\mathcal{L}_{n,\theta}(y) \leq -\frac{1}{2+\max_i|x_i^T(\theta-\theta_0)|}(\theta-\theta_0)^T X^T W X(\theta-\theta_0)$$

$$\leq -\frac{\kappa\|X\|^2\|\theta-\theta_0\|_2^2}{2+8s_0^{1/2}\|X\|_\infty\|\theta-\theta_0\|_2} =: -\frac{1}{2}r(\|\theta-\theta_0\|_2)$$

for the rate function $r(t) = \underline{\kappa}\|X\|^2 t^2/(1+4s_0^{1/2}\|X\|_\infty t)$. Thus the event $\check{\mathcal{E}}_{n,1}(\mathcal{N},r)$ holds deterministically true for any $y \in \{0,1\}^n$ and this choice of $r$. Furthermore, since $g''(t) \leq 1/4$, by considering the remainder in the Taylor expansion of $\ell_{n,\theta}(y) - \ell_{n,\theta_0}(y)$, for $\theta - \theta_0 \in \Theta_0 = \{\theta' : S_{\theta'} \subseteq S_{\theta_0}\}$,

$$\mathcal{L}_{n,\theta}(y) \geq -\tfrac{1}{8}(\theta-\theta_0)^T X^T X(\theta-\theta_0) \geq -\tfrac{1}{8}\bar{\kappa}(s_0)\|X\|^2\|\theta-\theta_0\|_2^2.$$

Thus $\hat{\mathcal{E}}_{n,1}(\Theta_0,\bar{L}) = \{0,1\}^n$ for $\bar{L} = \bar{\kappa}(s_0)\|X\|^2/4$. Inspection of the proof of Theorem 3 of [1] shows that he actually only requires the larger event $\mathcal{A}_{n,1}(\lambda/2) = \{\|\nabla_\theta\ell_{n,\theta_0}(Y)\|_\infty \leq \lambda/2\} \supsetneq$

$\mathcal{E}_{n,0}(\mathbb{R}^p, \lambda)$ to hold rather than $\mathcal{E}_{n,0}(\mathbb{R}^p, \lambda)$ (see p. 23 of [1] - they are incorrectly stated as being equal). In our setting, we may thus replace the event of Theorem 3(1) of [1] by

$$\mathcal{A}_{n,1}(\lambda/2) \cap \check{\mathcal{E}}_{n,1}(\mathcal{N}, r) \cap \hat{\mathcal{E}}_{n,1}(\Theta_0, \bar{L}) = \mathcal{A}_{n,1}(\lambda/2)$$

for $r, \bar{L}$ as above, from which the result follows.

Part (2): [1] considers the event $\mathcal{E}_{n,0}(\bar{\Theta}_L, \eta_L) \cap \hat{\mathcal{E}}_{n,1}(\Theta_0, \bar{L}) \cap \check{\mathcal{E}}_n(\bar{\Theta}_L, r)$, which we again simplify. From Part (1), we again take $\hat{\mathcal{E}}_{n,1}(\Theta_0, \bar{L}) = \{0,1\}^n$ for $\bar{L} = \bar{\kappa}(s_0)\|X\|^2/4$. Arguing as in Part (1), we get $\check{\mathcal{E}}_{n,1}(\bar{\Theta}_L, r) = \{0,1\}^n$ for rate function $r(x) = \frac{\kappa((L+1)s_0)\|X\|^2 x^2}{1 + \|X\|_\infty \sqrt{(L+1)s_0 x/2}}$ using that for $\theta \in \bar{\Theta}_L$, $|x_i^T(\theta - \theta_0)| \leq \|X\|_\infty \|\theta - \theta_0\|_1 \leq \|X\|_\infty \sqrt{(L+1)s_0}\|\theta - \theta_0\|_2$. For any $\theta \in \theta_0 + \bar{\Theta}_L$, by Cauchy-Schwarz,

$$|\langle \nabla_\theta \ell_{n,\theta_0}(y), \theta - \theta_0 \rangle| \leq \|\nabla_\theta \ell_{n,\theta_0}(y)\|_\infty \|\theta - \theta_0\|_1 \leq \|\nabla_\theta \ell_{n,\theta_0}(y)\|_\infty \sqrt{(L+2)s_0}\|\theta - \theta_0\|_2,$$

so that $\mathcal{E}_{n,0}(\bar{\Theta}_L, \eta_L) \supset \mathcal{A}_{n,1}(\|X\|\sqrt{\log p})$. We hence conclude that

$$\mathcal{E}_{n,0}(\bar{\Theta}_L, \eta_L) \cap \check{\mathcal{E}}_{n,1}(\bar{\Theta}_L, r) \cap \hat{\mathcal{E}}_{n,1}(\Theta_0, \bar{L}) = \mathcal{E}_{n,0}(\bar{\Theta}_L, \eta_L) \supset \mathcal{A}_{n,1}(\|X\|\sqrt{\log p})$$

for the above choices of $\bar{L}, r$ and $\eta_L$. Applying Theorem 3(2) of [1] then gives the result. $\square$

The following is the non-asymptotic analogue of Theorem 1 in Section 3.

**Theorem 4.** *Suppose the prior satisfies* (3) *and* (4), *and* $p^{A_4} \geq 8A_2$. *If for $K > 0$, the design matrix satisfies condition* (8) *with $L = L_K = \frac{K+C}{A_4 - \log(4A_2)/\log p}$ and $C$ the constant in Lemma 8, then for any $\theta_0 \in \mathbb{R}^p$,*

$$E_{\theta_0} Q^* \left( \theta \in \mathbb{R}^p : \|\theta - \theta_0\|_2 \geq C_K \frac{\sqrt{s_0 \log p}}{\|X\|} \right) \leq \frac{\zeta_n + 8e^{-(K/2)s_0 \log p}}{(K/2)s_0 \log p} + \frac{2}{p} + \frac{8}{3}p^{-s_0} + 8p^{-Ls_0},$$

*where $\zeta_n$ is given in Lemma 7, $C_K = \frac{8M_K\sqrt{L_K+2}}{\kappa((L_K+1)s_0)}$, and $M_K = \max(25\alpha, \frac{1+A_3}{16}, \frac{K+\tilde{C}_p}{8L_K} + \frac{1}{8} + \frac{\log(24)}{8\log p})$ with $\tilde{C}_p$ given in Lemma 9.*

*Furthermore, the mean-squared prediction error $\|p_\theta - p_0\|_n^2 = \frac{1}{n}\sum_{i=1}^n (\Psi(x_i^T\theta) - \Psi(x_i^T\theta_0))^2$ of the VB posterior $Q^*$ satisfies*

$$E_{\theta_0} Q^* \left( \theta \in \mathbb{R}^p : \|p_\theta - p_0\|_n^2 \geq \frac{C_K\sqrt{\bar{\kappa}((L_K+1)s_0)}}{4} \sqrt{\frac{s_0 \log p}{n}} \right)$$

$$\leq \frac{\zeta_n + 8e^{-(K/2)s_0 \log p}}{(K/4)s_0 \log p} + 2/p + (8/3)p^{-s_0} + 8p^{-Ls_0}.$$

*Proof.* We first apply Lemma 1 with

$$\Theta_n = \left\{ \theta \in \mathbb{R}^p : \|\theta - \theta_0\|_2 \geq C_K \frac{\sqrt{s_0 \log p}}{\|X\|} \right\},$$

$A = \mathcal{A}_n = \mathcal{A}_n(t, L, M_1, M_2)$ the event in Lemma 11, $\delta_n = Ks_0 \log p$, and $C = 8$. Since $\mathcal{A}_n \subset \mathcal{A}_{n,1}(\|X\|\sqrt{\log p})$, Lemma 10 implies that the condition (14) holds. Using Lemma 1 followed by Lemma 7,

$$E_{\theta_0} Q^* \left( \theta \in \mathbb{R}^p : \|\theta - \theta_0\|_2 \geq C_K \frac{\sqrt{s_0 \log p}}{\|X\|} \right) 1_{\mathcal{A}_n} \leq \frac{E_{\theta_0} \text{KL}(Q^* \| \Pi(\cdot|Y)) 1_{\mathcal{A}_n} + 8e^{-\delta_n/2}}{\delta_n/2}$$

$$\leq \frac{\zeta_n + 8\exp(-(K/2)s_0 \log p)}{(K/2)s_0 \log p}.$$

By Lemma 11,

$$E_{\theta_0} Q^* \left( \theta \in \mathbb{R}^p : \|\theta - \theta_0\|_2 \geq C_K \frac{\sqrt{s_0 \log p}}{\|X\|} \right) 1_{\mathcal{A}_n^c} \leq P_{\theta_0}(\mathcal{A}_n^c) \leq 2/p + (8/3)p^{-s_0} + 8p^{-Ls_0}.$$

The first statement follows by combining the above two displays.

Turning to the second statement, Lemma 10 implies the condition (14) holds with
$$\Theta_n = \{\theta \in \mathbb{R}^p : |S_\theta| \geq L_K s_0\},$$
$\delta_n = K s_0 \log p$ and $C = 2$. Therefore, similarly as above,
$$E_{\theta_0} Q^*\Big(\theta \in \mathbb{R}^p : |S_\theta| \geq L_K s_0\Big) 1_{\mathcal{A}_n} \leq \frac{\zeta_n + 2\exp(-(K/2)s_0 \log p)}{(K/2)s_0 \log p}.$$

For any $\theta$ in the set in the last display, since $\|\Psi'\|_\infty \leq 1/4$,
$$n\|p_\theta - p_0\|_n^2 \leq \frac{1}{16}\sum_{i=1}^n |x_i^T(\theta - \theta_0)|^2 \leq \frac{1}{16}\|X(\theta - \theta_0)\|_2^2 \leq \frac{1}{16}\bar{\kappa}((L_K+1)s_0)\|X\|^2\|\theta - \theta_0\|_2^2,$$

where the last inequality follows from the definition of $\bar{\kappa}(\cdot)$. The second statement then follows by combining the first statement of the theorem and the last two displays. $\qquad\square$

## 11  Deriving the variational algorithm

### 11.1  Coordinate ascent equations

Since the VB minimization problem (6) is intractable for Bayesian logistic regression, we instead minimize a surrogate objective obtained by lower bounding the likelihood [2, 10]. This is a standard approach, but we include full details for completeness. For the log-likelihood $\ell_{n,\theta}$ defined in (19), it holds that
$$\ell_{n,\theta}(x,y) \geq \sum_{i=1}^n \log \Psi(\eta_i) - \frac{\eta_i}{2} + (y_i - \tfrac{1}{2})x_i^T\theta - \frac{1}{4\eta_i}\tanh(\eta_i/2)\big((x_i^T\theta)^2 - \eta_i^2\big) =: f(\theta, \eta)$$
(30)

for any $\eta = (\eta_1, \ldots, \eta_n) \in \mathbb{R}^n$, see Section 11.2 for a proof. Hence for any distribution $Q$ for $\theta$,
$$
\begin{aligned}
\mathrm{KL}(Q\|\Pi(\cdot|Y)) &= \int \log\left(\frac{dQ(\theta)}{e^{\ell_{n,\theta}(x,y)}d\Pi(\theta)}\right)\, dQ(\theta) + C \\
&\leq \int \log\frac{dQ}{d\Pi}(\theta) - f(\theta, \eta)\, dQ(\theta) + C \\
&= \mathrm{KL}(Q\|\Pi) - E^Q[f(\theta, \eta)] + C,
\end{aligned}
$$
(31)

where $C$ is independent of $Q$. We minimize the right-hand side over the variational family $Q_{\mu,\sigma,\gamma} \in \mathcal{Q}$, i.e. over the parameters $\mu, \sigma, \gamma$. Since we seek the tightest possible upper bound in (31), we also minimize this over the free parameter $\eta$. In particular, the coordinate ascent variational inference (CAVI) algorithm alternates between updating $\eta$ for fixed $\mu, \sigma, \gamma$ and then cycling through $\mu_j, \sigma_j, \gamma_j$ and updating these given all other parameters are fixed.

Write $E_{\mu,\sigma,\gamma}$ for the expectation when $\theta \sim Q_{\mu,\sigma,\gamma}$. For fixed $\mu, \sigma, \gamma$, update $\eta = (\eta_1, \ldots, \eta_n)$ by
$$\eta_i^2 = E_{\mu,\gamma,\sigma}(x_i^T\theta)^2 = \sum_{k=1}^p \gamma_k x_{ik}^2(\mu_k^2 + \sigma_k^2) + \sum_{k=1}^p\sum_{l\neq k}(\gamma_k x_{ik}\mu_k)(\gamma_l x_{il}\mu_l),$$
(32)

see Section 11.2 for a proof. We now derive the coordinate update equations for $\mu_j, \sigma_j, \gamma_j$ keeping all other parameters, including $\eta$, fixed. For completeness, we allow the Laplace slab to have non-zero mean $\nu$ if desired.

**Proposition 1** (Coordinate updates with Laplace prior). *Consider the prior* (9) *with Laplace slab density* $g(x) = \frac{\lambda}{2}e^{-\lambda|x-\nu|}$, *where* $\nu \in \mathbb{R}$, $\lambda > 0$. *Given all other parameters are fixed, the values* $\mu_j$ *and* $\sigma_j$ *that minimize* (31) *with* $Q = Q_{\mu,\sigma,\gamma} \in \mathcal{Q}$ *are the minimizers of the objective functions:*

$$
\begin{aligned}
\mu_j \mapsto\ & \lambda\sigma_j\sqrt{\frac{2}{\pi}}e^{-\frac{(\mu_j-\nu)^2}{2\sigma_j^2}} + \lambda(\mu_j - \nu)\mathrm{erf}\left(\frac{\mu_j - \nu}{\sqrt{2}\sigma_j}\right) + \mu_j^2\sum_{i=1}^n \frac{1}{4\eta_i}\tanh(\eta_i/2)x_{ij}^2 \\
& + \mu_j\left(\sum_{i=1}^n \frac{1}{2\eta_i}\tanh(\eta_i/2)x_{ij}\sum_{k\neq j}\gamma_k x_{ik}\mu_k - \sum_{i=1}^n (y_i - 1/2)x_{ij}\right),
\end{aligned}
$$

$$
\sigma_j \mapsto\ \lambda\sigma_j\sqrt{\frac{2}{\pi}}e^{-\frac{(\mu_j-\nu)^2}{2\sigma_j^2}} + \lambda(\mu_j - \nu)\mathrm{erf}\left(\frac{\mu_j - \nu}{\sqrt{2}\sigma_j}\right) - \log\sigma_j + \sigma_j^2\sum_{i=1}^n \frac{1}{4\eta_i}\tanh(\eta_i/2)x_{ij}^2,
$$

*respectively, where* $\mathrm{erf}(x) = 2/\sqrt{\pi} \int_0^x e^{-t^2} dt$ *is the error function. The value* $\gamma_j$ *that minimizes* (31) *given all other parameters are fixed, is the solution to*

$$-\log\frac{\gamma_j}{1-\gamma_j} = \log\frac{b_0}{a_0} - \log(\lambda\sigma_j) + \lambda\sigma_j\sqrt{\frac{2}{\pi}}e^{-\frac{(\mu_j-\nu)^2}{2\sigma_j^2}} + \lambda(\mu_j-\nu)\mathrm{erf}\left(\frac{\mu_j-\nu}{\sqrt{2}\sigma_j}\right) - \frac{1}{2}$$

$$-\mu_j\sum_{i=1}^n(y_i-1/2)x_{ij} + \sum_{i=1}^n\frac{1}{4\eta_i}\tanh(\eta_i/2)\left(x_{ij}^2(\mu_j^2+\sigma_j^2) + 2x_{ij}\mu_j\sum_{k\neq j}\gamma_k x_{ik}\mu_k\right).$$

*Proof.* Throughout this proof we fix the parameter $\eta \in \mathbb{R}^n$ and let $C$ denote any term constant with respect to the parameters currently being optimized, possibly different on each line. We first compute the update equations for $\mu_j$ and $\sigma_j$ based on (31). We compute $\mathrm{KL}(Q_{\mu,\sigma,\gamma|z_j=1}||\Pi)$, which considers the distribution $Q_{\mu,\sigma,\gamma}$ conditional on $z_j = 1$, as a function of $(\mu_j, \sigma_j)$, holding all other parameters fixed. Using that $Q_{\mu,\sigma,\gamma}$ is a factorizable distribution and that conditional on $z_j = 1$ the variational distribution of $\theta_j$ is singular to the Dirac measure $\delta_0$, we can simplify $\frac{dQ_{\mu,\sigma,\gamma|z_j=1}}{d\Pi} = C\frac{dQ_{\mu_j,\sigma_j|z_j=1}}{d\Pi_j^c}$, where $\Pi_j^c$ is the continuous part of the prior distribution for $\theta_j$ and $C$ does not depend on $\mu_j$ or $\sigma_j$. Recall that $\Pi_j^c = \int_0^1 w\mathrm{Lap}(\nu,\lambda)dw = a_0/(a_0+b_0)\mathrm{Lap}(\nu,\lambda)$. Thus for $\phi_{\mu,\sigma}$ the density of a $N(\mu,\sigma^2)$ distribution and $\overline{w} = a_0/(a_0+b_0)$,

$$\log\frac{dQ_{\mu,\sigma,\gamma|z_j=1}}{d\Pi}(\theta) = \log\frac{dQ_{\mu_j,\sigma_j|z_j=1}}{d\Pi_j^c}(\theta_j) + C = \log\frac{\phi_{\mu_j,\sigma_j}(\theta_j)}{\overline{w}g_j(\theta_j)} + C.$$

Taking expectations with respect to $Q_{\mu,\sigma,\gamma|z_j=1}$,

$$E_{\mu,\sigma,\gamma|z_j=1}\left[\log\frac{\phi_{\mu_j,\sigma_j}(\theta_j)}{\overline{w}g_j(\theta_j)}\right] = E_{\mu,\sigma|z_j=1}\left[-\log(\lambda\sigma_j) - \frac{(\theta_j-\mu_j)^2}{2\sigma_j^2} + \lambda|\theta-\nu|\right] + C$$

$$= -\log(\lambda\sigma_j) + \lambda E_{\mu,\sigma|z_j=1}|\theta_j-\nu| + C,$$

where we have used $E_{\mu,\sigma,\gamma|z_j=1}[(\theta_j-\mu_j)^2/\sigma_j^2] = 1$. Under the variational distribution, $\lambda|\theta_j-\nu|$ follows a folded Gaussian distribution, hence

$$\lambda E_{\mu_j,\sigma_j|z_j=1}|\theta_j-\nu| = \lambda\sigma_j\sqrt{\frac{2}{\pi}}e^{-\frac{(\mu_j-\nu)^2}{2\sigma_j^2}} + \lambda(\mu_j-\nu)\mathrm{erf}\left(\frac{\mu_j-\nu}{\sqrt{2}\sigma_j}\right).$$

Combining the last three displays gives $\mathrm{KL}(Q_{\mu,\sigma,\gamma|z_j=1}||\Pi)$ as a function of $\mu_j, \sigma_j$. Using this expression and evaluating $E_{\mu,\sigma,\gamma|z_j=1}\big[f(\theta,\eta)\big]$ using Lemma 15 below, the upper bound in (31) equals, as a function of $\mu_j, \sigma_j$,

$$-\log(\lambda\sigma_j) + \lambda\sigma_j\sqrt{\frac{2}{\pi}}e^{-\frac{(\mu_j-\nu)^2}{2\sigma_j^2}} + \lambda(\mu_j-\nu)\mathrm{erf}\left(\frac{\mu_j-\nu}{\sqrt{2}\sigma_j}\right)$$

$$-\sum_{i=1}^n(y_i-1/2)x_{ij}\mu_j + \sum_{i=1}^n\frac{1}{4\eta_i}\tanh(\eta_i/2)\left(x_{ij}^2(\mu_j^2+\sigma_j^2) + 2x_{ij}\mu_j\sum_{k\neq j}\gamma_k x_{ik}\mu_k\right) + C,$$

where $C$ is independent of $\mu_j, \sigma_j$. Minimizing the display with respect to either $\mu_j$ or $\sigma_j$ gives the desired result.

For updating the inclusion probabilities $\gamma_j$, we proceed as above without conditioning on $z_j = 1$. Keeping track of only the $\gamma_j$ terms,

$$E_{\mu,\sigma,\gamma}\left[\log\frac{dQ_{\mu,\sigma,\gamma}}{d\Pi}(\theta)\right] = E_{\mu,\sigma,\gamma}\left[\log\frac{d(\gamma_j N(\mu_j,\sigma_j^2) + (1-\gamma_j)\delta_0)}{d(\overline{w}\mathrm{Lap}(\nu,\lambda) + (1-\overline{w})\delta_0)}(\theta_j)\right] + C$$

$$= E_{\mu,\sigma,\gamma}\left[1_{\{z_j=1\}}\log\frac{\gamma_j dN(\mu_j,\sigma_j^2)}{\overline{w}d\mathrm{Lap}(\nu,\lambda)}(\theta_j) + 1_{\{z_j=0\}}\log\frac{1-\gamma_j}{1-\overline{w}}\right] + C$$

$$= \gamma_j E_{\mu,\sigma,\gamma|z_j=1}\left[\log\frac{\phi_{\mu_j,\sigma_j}(\theta_j)}{g_j(\theta_j)}\right] + \gamma_j\log\frac{\gamma_j}{\overline{w}} + (1-\gamma_j)\log\frac{1-\gamma_j}{1-\overline{w}} + C.$$

The first expectation was evaluated above. Using this and evaluating $E_{\mu,\sigma,\gamma}\big[f(\theta,\eta)\big]$ using Lemma 15 below, the upper bound in (31) equals, as a function of $\gamma_j$,

$$\gamma_j\Bigg\{-\log(\lambda\sigma_j)+\lambda\sigma_j\sqrt{\frac{2}{\pi}}e^{-\frac{(\mu_j-\nu)^2}{2\sigma_j^2}}+\lambda(\mu_j-\nu)\mathrm{erf}\left(\frac{\mu_j-\nu}{\sqrt{2}\sigma_j}\right)-\mu_j\sum_{i=1}^{n}(y_i-1/2)x_{ij}$$

$$-\frac{1}{2}+\sum_{i=1}^{n}\frac{1}{4\eta_i}\tanh(\eta_i/2)\left(x_{ij}^2(\mu_j^2+\sigma_j^2)+2x_{ij}\mu_j\sum_{k\neq j}\gamma_k x_{ik}\mu_k\right)\Bigg\}$$

$$+\gamma_j\log\frac{\gamma_j}{\varpi}+(1-\gamma_j)\log\frac{1-\gamma_j}{1-\varpi}+C,$$

where $C$ is independent of $\gamma_j$. As a function of $\gamma_j$, this takes the form

$$h(\gamma_j)=\gamma_j\log\frac{\gamma_j}{a}+(1-\gamma_j)\log\frac{1-\gamma_j}{b}+c\gamma_j,$$

with $a,b\in(0,1)$ and $c\in\mathbb{R}$. By differentiating, $h$ is convex and has a global minimizer $\bar\gamma_j\in[0,1]$ satisfying

$$-\log\frac{\bar\gamma_j}{1-\bar\gamma_j}=c+\log\frac{b}{a}.$$

Substituting in the above values for $a,b,c$ gives the result. $\qquad\square$

## 11.2  Variational lower bound

We now derive the lower bound (30) as in [10]. Recall the log-likelihood (19):

$$\ell_{n,\theta}(x,y)=\sum_{i=1}^{n}y_i x_i^T\theta-g(x_i^T\theta),$$

where $g(t)=\log(1+e^t)$, $t\in\mathbb{R}$. We lower bound the second term above using a Taylor expansion in $x^2$. The following lemma fills in some details from [10], where this technique was proposed.

**Lemma 13.** *For $\Psi(x)=(1+e^{-x})^{-1}$ the standard logistic function and any $\eta\in\mathbb{R}$,*

$$\log\Psi(x)\geq\frac{x-\eta}{2}+\log\Psi(\eta)-\frac{1}{4\eta}\tanh(\eta/2)(x^2-\eta^2).$$

*Proof.* Note that we can write $\log\Psi(x)=x/2-\log(e^{x/2}+e^{-x/2})$. By elementary calculations, it can be shown that the second term on the right hand side is convex in the variable $x^2$. We can therefore use its first order Taylor approximation in $x^2$ to derive a lower bound for $\log\Psi(x)$, i.e. for any $\eta\in\mathbb{R}$,

$$\log\Psi(x)\geq\frac{x}{2}-\log(e^{\eta/2}+e^{-\eta/2})-\frac{1}{4\eta}\tanh(\eta/2)(x^2-\eta^2)$$

$$=\frac{x-\eta}{2}+\log\Psi(\eta)-\frac{1}{4\eta}\tanh(\eta/2)(x^2-\eta^2).$$

$\qquad\square$

Substituting this lower bound into each term $-g(x_i^T\theta)=\log\Psi(-x_i^T\theta)$ in the log-likelihood yields (30).

We next obtain the update equation (32) for the free parameter $\eta\in\mathbb{R}^n$. Minimizing (31) over $\eta$ for fixed $Q=Q_{\mu,\sigma,\gamma}\in\mathcal{Q}$ is equivalent to solving

$$\widetilde\eta=\arg\max_{\eta\in\mathbb{R}^n}E_{\mu,\sigma,\gamma}\big[f(\theta,\eta)\big].$$

**Lemma 14.** *The function $f_a:\mathbb{R}\to\mathbb{R}$, $a\geq0$, given by*

$$f_a(x)=\log\Psi(x)-\frac{x}{2}-\frac{1}{4x}\tanh(x/2)\big(a^2-x^2\big)$$

*is symmetric about zero and possesses unique maximizers at $x=\pm a$.*

*Proof.* Indeed, $\tanh(-x) = -\tanh(x)$ shows that the last term is symmetric around zero. Moreover, $\Psi(-x) = 1 - \Psi(x)$ yields $\log \Psi(-x) + x/2 = \log \Psi(x) - x/2$ thereby proving symmetry of $f_a$. Using $2\Psi(x) - 1 = \tanh(x/2)$,

$$f_a'(x) = \frac{\Psi'(x)}{\Psi(x)} - \frac{1}{2} + \frac{1}{4}\left(\tanh(x/2) + \frac{x}{2(\cosh x/2)^2}\right) - \left(\frac{x(\cosh x/2)^{-2} - 2\tanh(x/2)}{8x^2}\right)a^2$$

$$= (a^2 - x^2)\left(\frac{\tanh(x/2)}{4x^2} - \frac{1}{8x(\cosh x/2)^2}\right).$$

Since $f_a'(\pm a) = 0$, it remains to show $f_a''(\pm a) < 0$. Note that

$$f_a''(x) = (a^2 - x^2)\frac{d}{dx}\left(\frac{\tanh(x/2)}{4x^2} - \frac{1}{8x(\cosh x/2)^2}\right) - \frac{\tanh(x/2)}{2x} + \frac{1}{4(\cosh x/2)^2}.$$

The first term vanishes at $x = \pm a$ and the second is symmetric about zero, so the formula $\sinh(x)\cosh(x) = \sinh(2x)/2$ yields

$$f_a''(\pm a) = -\frac{2\sinh(a/2)\cosh(a/2) - a}{4a(\cosh a/2)^2} = -\frac{\sinh(a) - a}{4a(\cosh a/2)^2}.$$

This concludes the proof as $\sinh(x)/x \geq 1$. $\qquad\square$

By the last lemma, we can restrict the free parameter to $\eta \in \mathbb{R}_{\geq 0}^n$ and take $\widetilde{\eta}_i^2 = E_{\mu,\gamma,\sigma}(x_i^T\theta)^2$ to maximize $E_{\mu,\sigma,\gamma}f(\theta,\eta)$. The update (32) then follows from the following lemma.

**Lemma 15.** *For* $Q_{\mu,\sigma,\gamma} \in \mathcal{Q}$,

$$\mathbb{E}_{\mu,\sigma,\gamma}[x_i^T\theta] = \sum_{k=1}^{p}\gamma_k\mu_k x_{ik},$$

$$\mathbb{E}_{\mu,\sigma,\gamma}\left[(x_i^T\theta)^2\right] = \sum_{k=1}^{p}\gamma_k x_{ik}^2(\mu_k^2 + \sigma_k^2) + \sum_{k=1}^{p}\sum_{l\neq k}(\gamma_k x_{ik}\mu_k)(\gamma_l x_{il}\mu_l).$$

*If the expectations are instead taken over* $E_{\mu,\sigma,\gamma|z_j=1}$, *then the same formulas hold true with* $\gamma_j = 1$.

*Proof.* Since $\theta_k \sim^{iid} (1 - \gamma_k)\delta_0 + \gamma_k\mathcal{N}(\mu_k,\sigma_k^2)$ under $Q_{\mu,\sigma,\gamma}$, the first claim follows by linearity of the expectation. Using that $\theta_k = \theta_k 1_{\{z_k=1\}}$, $Q_{\mu,\sigma,\gamma}$-almost surely, and that $(\theta_k)$ are independent under the mean-filed distribution $Q_{\mu,\sigma,\gamma}$,

$$E_{\mu,\sigma,\gamma}\left[(x_i^T\theta)^2\right] = E_{\mu,\sigma,\gamma}\left(\sum_{k=1}^{p}x_{ik}\theta_k 1_{\{z_k=1\}}\right)^2$$

$$= \sum_{k=1}^{p}\gamma_k x_{ik}^2(\mu_k^2 + \sigma_k^2) + \sum_{k=1}^{p}\sum_{l\neq k}(\gamma_k x_{ik}\mu_k)(\gamma_l x_{il}\mu_l).$$

$\qquad\square$