[Reviews · NeurIPS 2020]

Review 1

Summary and Contributions: This paper established optimal bounds for VB in a high-dimensional sparse logistic regression model and proposed a VB algorithm that was empirically shown by the authors to be an appealing alternative to the existing procedures.

Strengths: Disclaimer first: Bayesian inference is not in my area, so my evaluation is based on an apparent ignorance of the large literature along this very interesting line of research. To me, the theoretical bounds established in this paper are meaningful and interesting, and if they are new, they are to me of added value. One potential concern is how much more work has to be done give what has been known regarding linear regression, about which I believe much has been understood. However, since I did not do much related to VB, I will leave this point to other referees as well as the AC to comment on. A new VB algorithm is also proposed and empirically shown to be performing better than the existing ones. I am not sure how much novelty there is though; the algorithm looks standard to me.

Weaknesses: It seems that the authors did not analyze the convergence of their algorithm. I would suggest the authors to comment on the theoretical validity of their algorithm.

Correctness: They read meaningful to me.

Clarity: It is well written to me.

Relation to Prior Work: I think so; but again, VB is not in my area.

Reproducibility: Yes

Additional Feedback: Restricted to the studied problem, I would love to see more comments on the advantage of VB over frequentist approaches using, say, penalized MLE. It is my understanding that the main advantage of VB is not on estimation/prediction but on inference (e.g., establishing confidence intervals)? If so, would establishing validity of the confidence interval derived by VB (i.e., Bernstein-von Mises type results) be more interesting? [Update after rebuttal] I really appreciate the authors' comments on my questions. They are exceedingly clear to me, and combined with the other referees' comments on novelty, made me to accordingly raise my score further. Speaking about Bernstein-von Mises type results, in case the authors missed it, V. Spokoiny had some very exciting progresses to extend them to high dimensions in a general M-estimation framework; cf. https://projecteuclid.org/euclid.ba/1422884986. Of course, I believe they are still millions miles away from being applicable to studying VB, but maybe useful in terms of strengthening the results of Wang and Blei (?). I sincerely hope that the authors continue their success in this rather exciting line of research!!!


Review 2

Summary and Contributions: The paper aims to provide statistical guarantees for variational Bayes (VB) method when a high-dimensional logistic regression model is under consideration. The authors show that under an appropriate prior (spike and slab with Laplace slabs), VB can achieve minimax rate under both \ell_2 and mean-squared prediction loss. A coordinate-ascent variational inference algorithm is introduced to compute the VB posterior. Several numerical results are presented to verify the theoretical results. In particular, both \ell_2 and mean-squared prediction loss are under control, and the VB posterior can also control FDR in terms of variable selection.

Strengths: The main contribution of the paper is to derive the minimax concentration rate for the VB Posterior when a high-dimensional logistic regression model is considered. Theoretical guarantees of VB have drawn a lot of attention in recent years. The paper relies on the recent breakthrough work on this topic, but also has its own contributions.

Weaknesses: I think several key reference papers are not cited in the paper. This includes the Skinny Gibbs (Narisetty et. al., 2019, JASA), which shows strong variable selection consistency under a spike and slab prior for high-dimensional logistic regression models and propose an efficient algorithm for sampling; and the paper by Ran Wei and Subhashis Ghosal, 2019, which obtains minimax posterior contraction rates for high-dimensional logistic regression models under a wide class of shrinkage priors. In simulation part, I think it would be more convincing if the authors can also include a comparison with the Skinny Gibbs.

Correctness: I did not check the details of the proof. It seems that the outline is correct.

Clarity: The paper is well written and easy to follow.

Relation to Prior Work: As mentioned before, I think several key reference papers are missing.

Reproducibility: Yes

Additional Feedback: -------------------------------------------------- -------after reading authors' feedback------- The reason why I say Wei and Ghosal's paper is missing is that, when the authors review the relevant work this paper is not mentioned. It is only mentioned to explain technical assumptions. I also hope the authors can add the comparison with the skinny Gibbs. My score remains the same as before.


Review 3

Summary and Contributions: The rebuttal addressed well my concern on the use of surrogate KL and to some extent the comment on the practical relevance. Hence, I have raised my score to 7. The paper establishes some theoretical guarantees for variable selection with a spike and slab prior and mean-field VB approximation. This is a highly relevant topic given the recent renewed interest in this area.

Strengths: The paper has a solid theoretical grounding. I only skimmed over the proofs, but they all sound correct and carefully written.

Weaknesses: The main limitation is its practical relevance. The CAVI algorithm for minimizing the original KL divergence is challenging to derive, an alternative is developed that minimizes a surrogate KL in Equation (10). Some explanation and motivation about using this surrogate should be provided; also, it'd be great if the authors can give some comments on efficiency of this surrogate target.

Correctness: Yes

Clarity: Yes

Relation to Prior Work: Yes

Reproducibility: Yes

Additional Feedback: line 25 "explainability and interpretability": what is the difference between these two terms?


Review 4

Summary and Contributions: The authors show some non-asymptotic theoretical guarantees for the Variational Bayes algorithm and illustrate the improvement in performance of the algorithm relative to sparse VB approaches. The results highlight that the variational approximation using Laplace slabs outperforms the VB method with Gaussian prior slab available in the literature.

Strengths: The strength of the article is the theoretical guarantees results obtained for optimal concentration of VB posterior. VB approaches are shown to be faster than other approaches though the gain in accuracy is not at the same scale.

Weaknesses: It is claimed that their approach can be used in high-dimensional models where other approaches based on the EM algorithm or MCMC are not computable. Though I agree with this, I think it's better to have some application based support for this by applying the methods on large models and do a model assessment. Also, a discussion on how sensitive the results in table 1 with respect to settings of hyper parameters will enhance the quality of the results.

Correctness: The results seem to be correct according to my knowledge.

Clarity: Yes, the paper is well written and some clarifications can be added on experimental settings.

Relation to Prior Work: The paper seems to be fairly acknowledge the prior work available in the literature.

Reproducibility: Yes

Additional Feedback:

[Author Response · NeurIPS 2020]

We thank the reviewers for their constructive suggestions. We have (1) added simulations for (i) Skinny Gibbs, (ii)
different hyperparameter choices, (iii) coverage of our VB credible sets, and (2) expanded the discussion in the final
version to address the reviewer comments. A summary of the added discussion is provided point-by-point below.

**Reviewer 1: How much more work is needed versus linear regression.** In linear regression, one can do exact
computations using the Gaussian likelihood to yield precise oracle results [Ray and Szabo (2019)]. Such exact
expressions are not available for logistic regression, so we must instead use a different test-based proof using general
ideas from Bayesian nonparametrics (Section 10). The technical details are thus different (and more involved) here.

**Novelty of the VB algorithm.** A methodological novelty here is using Laplace slabs for the *prior* underlying the VB
approximation, rather than Gaussian slabs as all previous works do, and we show this does better empirically (Sections
5 & 8). We agree that deriving the resulting CAVI algorithm is somewhat standard, but the Laplace slabs modify the
usual Gaussian update equations and are needed for implementation/simulations. We emphasize our main contribution
is to provide theoretical guarantees and show our VB calibration empirically outperforms existing (Gaussian) VB
approaches, rather than novelty of the optimization algorithm. We have included these derivations for completeness.

**Comment on the theoretical validity of the algorithm.** To the best of our knowledge, convergence properties of
CAVI is still largely an open problem in even simple models, see e.g. Plummer et al. (July 2020). It is empirically known
that the VB optimization problem is typically difficult and non-convex, and that CAVI will not return a global optimizer.
However, with proper initialization, one still often recovers a good VB posterior (as we see in our simulations). This is
an excellent point, but unfortunately well beyond the scope of our paper.

**Comment on the advantage of VB over frequentist approaches.** A main advantage is access to variable inclusion
probabilities and their credible sets, which are often as equally interesting to practitioners as estimation. Our VB
approach performs well empirically for model selection, as demonstrated by its good FDR, TPR and coverage of
credible sets, which we have now added to the simulations. We have also added discussion on this point, thank you.

**Establishing the frequentist validity of VB credible intervals (i.e. Bernstein-von Mises type results).** Bernstein-
von Mises results have only been proved for VB in low-dimensional settings [Wang and Blei (2019)], where one can
modify classical local asymptotic normality (LAN) arguments for parametric models. Since LAN expansions do not
generally hold for high-dimensional models, including logistic regression, new proof techniques are required.

**Reviewer 2: Missing references and MCMC method.** We have added the missing reference for Skinny Gibbs
[Narisetty et al. (2019)] and have added the method in our simulations. Skinny Gibbs is indeed an order of magnitude
faster than MCMC using Stan, but generally 50-100 times slower than the VB methods. It provided broadly similar
FDR and TPR as VarBVS. Wei and Ghosal (2019) was already cited in the manuscript. Thank you for this suggestion.

**Reviewer 3: Practical relevance.** While this is a general issue with theory, it can often inform practice. Many
methodology/applied researchers are unaware that using light tailed (e.g. Gaussian) prior slabs can yield poor inference
for *true* sparse Bayesian inference, while the situation is even more complicated for VB. Indeed, we are unaware of
any existing VB papers for logistic regression *not* using prior Gaussian slabs. It is practically important to pick heavy
enough slabs and our theory confirms that exponential tails (Laplace) are sufficient for estimation when using VB.
These findings are fully reflected in practice, where our use of Laplace slabs consistently and significantly outperforms
the usual Gaussian slab approach in almost all simulations (Sections 5 & 8).

Our theory also provides conditions on the design matrix, which include many common examples, under which sparse
VB works. The non-asymptotic nature of our full results (Section 10) also confirm these lessons apply for reasonable
sample sizes, as demonstrated by our simulations. While our contribution is clearly on the more theoretical side, we
think the routine use of VB in machine learning, including for logistic regression, and the practical insights afforded by
our results, mean a machine learning conference is the right venue for our work.

**Explanation and efficiency of using the surrogate KL for CAVI.** The use of a surrogate KL functional (arising from
maximizing a lower bound on the marginal likelihood) is a standard technique for VB in Bayesian logistic regression,
see e.g. Chapter 10.6 of the textbook Bishop (2006). Its performance and motivation have been studied in several
papers, which we now cite more clearly [including Bishop (2006)]. We agree that we were too quick on this point and
have expanded the explanation, as well as providing references to more extensive discussions. Thank you.

**Reviewer 4: Add application based support with a large model and do a model assessment.** It is known that for
large variable selection problems, MCMC methods often mix poorly and we should not assume MCMC estimates are
close to exact values [Carbonetto and Stephens (2015), Griffin et al. (2017)]. Hence VB methods have been extensively
used in the literature. If required, we can add a large real-world dataset with several thousand features in the supplement.

**Discuss sensitivity to hyperparameter selection.** We have added a simulation study on this: we find that the choice of
hyperparameter does indeed affect the small-sample behaviour, which we now report/discuss. Thank you for this point.

[Meta-Review · NeurIPS 2020]

This paper seems a solid theoretical contribution to the area of Variational Bayes, and most of the the reviewers concerns were addressed satisfactorily in the rebuttal, provided the mentioned simulations (particularly vs Skinny Gibbs) and comparisons are included in the final version. We hope that the authors incorporate their rebuttal into the final version, and expand the related work section.